DOI: 10.1038/s41467-017-02072-4　　**OPEN**

# Functional reduction in pollination through herbivore-induced pollinator limitation and its potential in mutualist communities

Paul Glaum [1] & André Kessler[2]

Plant–pollinator interactions are complex because they are affected by both interactors' phenotypes and external variables. Herbivory is one external variable that can have divergent effects on the individual and the population levels depending on specific phenotypic plastic responses of a plant to herbivory. In the wild tomato, *Solanum peruvianum*, herbivory limits pollinator visits, which reduces individual plant fitness due to herbivore-induced chemical defenses and signaling on pollinators (herbivore-induced pollinator limitation). We showed these herbivory-induced decreases in pollination to individual plants best match a Type II functional-response curve. We then developed a general model that shows these individual fitness reductions from herbivore-induced changes in plant metabolism can indirectly benefit overall populations and community resilience. These results introduce mechanisms of persistence in antagonized mutualistic communities that were previously found prone to extinction in theoretical models. Results also imply that emergent ecological dynamics of individual fitness reductions may be more complex than previously thought.

---

[1] Department of Ecology and Evolutionary Biology, University of Michigan, 830 North University, Ann Arbor, MI 48109, USA. [2] Department of Ecology & Evolutionary Biology, Cornell University, Ithaca, NY 14853, USA. Correspondence and requests for materials should be addressed to P.G. (email: prglaum@umich.edu)

Plant–animal interactions are inherently complex because they are affected by the phenotypes of the interacting species and the environment in which the interactions play out[1]. In an attempt to reduce this complexity and develop tractable questions, numerous studies of plant–animal interactions have focused on "single-interactions" such as herbivory, predation (carnivorous plants), seed dispersal, habitation-mutualisms, or pollination[2]. Such two-dimensional studies have provided much of our mechanistic understanding of species interactions, but provide only a limited picture of the ecology and evolution of plant–animal interactions[3]. In particular, herbivory-induced changes in plant secondary metabolism have been found to mediate complex dynamics in interaction networks by affecting the suitability of a host plant to other herbivore species[4–6], as well as the attraction of third[7] and fourth trophic level predators and parasitoids[8] with complex effects on plant fitness. Particularly interesting in this context is the plant metabolism-mediated interaction between herbivores and pollinators, because it is here where plants are exposed to a conflict of attracting mutualists (i.e., pollinators) and repelling antagonist consumers (i.e., herbivores) of plant tissues, using similarly structured chemical information[1].

Herbivory can affect plant pollinator interactions in multiple ways[9]. Reduction in pollinator visitation can result from altered/damaged floral displays[10–13] or pollinators actively avoiding contact with herbivores on flowers[14]. Moreover, herbivore attack usually results in plant metabolic changes that can affect the quality and quantity of pollinator rewards (either nectar or pollen)[15–17] or the chemical information that is mediating the interactions[18,19].

Particularly important in this respect, are herbivory-induced volatile organic compounds (HI-VOCs) that are emitted by plants in response to herbivory and provide a cue about the plants' metabolic state and chemical defense status. This form of chemical information can attract natural enemies (predators and parasitoids) of herbivores, mediate interactions with herbivores[7] or induce preemptive resistance in neighboring branches and plants[20]. It was hypothesized that the production of this chemical information can allow plants to manipulate the entire interaction network to minimize the impact of antagonistic interactions such as herbivory[1,21]. However, multi-functionality and ubiquitous availability of chemical information in general and HI-VOC emission in particular can become problematic for the plant if the same information is mediating interactions between antagonists and mutualists of the plant. In particular, if antagonists (e.g., herbivores) and mutualists (e.g., pollinators) both consume plant tissue or metabolites and use the associated chemical information for host choice[18,19,22].

In one example, the wild tomato Solanum peruvianum, herbivore-induced changes in plant metabolism and herbivore-induced volatile organic compounds (HI-VOC)-mediated information transfer reduces the attraction of bee pollinators to herbivore-attacked plants relative to undamaged plants. This negatively affected plant fitness via reduced pollen deposition when measured in the field[18,19]. Such interactions have been termed herbivory-induced pollinator limitation (HIPL), whereby indirect plant trait-mediated effects negatively affect interactions with a mutualist species and so reduce fitness of an individual plant. However, the broader effects on population and community dynamics and persistence of plant-induced responses, such as HIPL, have not been investigated. Herbivore-induced changes to plant metabolism, i.e., through HI-VOCs, alter how pollinators interact with flowering plants, which can be predicted to alter population dynamics and the dynamics of other interacting species within the community. Here we propose a data-driven theoretical model-based approach to address higher level effects of HIPL.

Theoretical models of the three species community flowering plant, pollinator, and herbivore (3-dimensions) have moved beyond single-interaction studies and investigated the direct effects of herbivory on mutualist populations[23,24]. However, many have not included indirect trait-mediated effects in their analyses. For example, considering only the direct effect of herbivory reducing plant population abundance, some of the model-based studies have concluded that these 3-dimensional systems are dissipative so the mutualism is prone to extinction unless herbivore attack rates and/or efficiencies are kept low[25–27]. In general, the extinctions predicted by these models are triggered by herbivores directly reducing plant population abundances and growth. As herbivory reduces the actual plant population size, this limits the amount of resources available to the pollinator population and causes a subsequent reduction in the pollinator population. The smaller abundance of pollinators reduces pollination services and then lowers plant reproduction, starting a feedback loop that can further reduce both mutualist populations to local extinction. These models have found extinction to be especially likely when the mutualism is an obligate or highly specialized mutualism, where each mutualist species is fairly dependent on the other for substantial growth[26,27].

Unlike direct herbivory, herbivore-induced pollinator limitation (HIPL) is an indirect effect. HIPL does not directly reduce the actual plant population size but does lower the rate of interactions between existing plants and pollinators. In other words, HIPL can reduce the effective population of plants the pollinators interact with as a function of the strength of induced plant metabolic changes in response to herbivory. Such an indirect ecological effect mediated by herbivore-induced changes in plant metabolism will also reduce pollination services and can thus be predicted to similarly induce mutualist extinction as had been found in previous models[25–27]. However, we show that the inclusion of mechanisms like HIPL into models generates the potential for unexpected population and community level effects that can reduce the tendency for extinction and actually support community persistence.

We generate this model using the empirical data of the effects of herbivory on pollination through HI-VOCs[18] (see Methods section). This data set measured HI-VOC release and pollinator visits at different levels of herbivory to determine how bee pollination of wild tomato plants changes as a function of the amount of herbivory experienced by a plant. Since pollinator visits change as a function of the level of herbivory, we call the resulting change in pollination the "functional form of HIPL." This functional form of HIPL can then be inserted into dynamic models of a flowering plant–pollinator–herbivore community to ascertain its effects on community dynamics and persistence with different pollinator relationships.

There are three objectives to the research presented here. (1) Find the functional form of HIPL by determining how pollinator visitation declines as a function of herbivory intensity. (2) Measure the effects of HIPL on the persistence of the interacting community and its dynamics through time at different rates of herbivory. (3) Compare the effects of HIPL on community dynamics and persistence in both obligate/highly specialized and facultative/generalist pollination relationships.

## Results

**Best fit functional form of HIPL.** We first establish the functional form that reduced pollinator attraction takes in relation to increased herbivory/herbivore presence using Kessler et al.'s data[18] (see Methods section). The function describing pollinator visitation decline will be denoted as $v(c, h)$ where $h$ is the percentage of herbivore-damaged leaves on a plant and $c$ is a

| Fitted models to averages | Example of functional form | Estimated parameters | Significance and fit | AICc | AICc weight |
|---|---|---|---|---|---|
| 1.)Type I/Linear: $\sim ch + i$ | | $c = -0.635$, $i = 0.8615$ | $p = (c)\ 3.05e^{-6}$ $(i)\ 1.29e^{-9}$ $R^2 = 0.9114$ | −22.262 | 0.02995 |
| 2.)Type II: $\sim \dfrac{i}{1 + ch}$ | | $c = 1.877$, $i = 0.9565$ | $p = (c)\ 9.87e^{-6}$ $(i)\ 8.74e^{-10}$ | −28.649 | 0.73011 |
| 3.)Type III: $\sim \dfrac{i}{1 + ch^2}$ | | $c = 2.418$ $i = 0.8419$ | $p = (c)\ 2.12e^{-4}$ $(i)\ 2.84e^{-9}$ | −21.524 | 0.02071 |
| 4.)Mixed Saturating: $\sim \dfrac{i}{1 + ch^b}$ | | $c = 2.022$ $b = 1.230$ $i = 0.9240$ | $p = (c)\ 2.71e^{-5}$ $(b)\ 0.000111$ $(i)\ 1.19e^{-8}$ | −26.240 | 0.21889 |
| 5.)Concave: $\sim i * \left(1 - \dfrac{h}{1}\right)^{c}$ | | $c = 0.5394$, $i = 0.8278$ | $p = (c)0.00045$ $(i)\ 7.41e^{-10}$ | −13.279 | 0.00034 |

**Fig. 1** Curve fitting the five candidate response models. Describing the results of the curve fitting to the five candidate response models for $v(c, h)$: Type I/ Linear, Type II, Type III, Mixed Saturating, Concave. Here $h$ represents the level of herbivory. The parameters $c$ and $b$ determine the shape of the curve and $i$ is the intercept. Equation representations of each model are given along with a pictorial example of each model. The Type II functional response has the highest Akaike Information Criterion weight of 0.73011

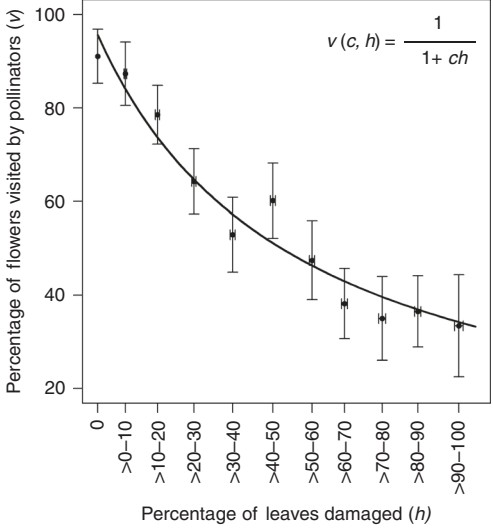

**Fig. 2** Best fit Type II functional form. Best fit Type II functional response of pollinator visitation ($v(c, h)$) as a function of proportionate leaf damage ($h$). Error bars show standard error of the mean

theoretical literature[28], their shown applicability in other interactions (such as predation and mutualist interaction)[29,30], and their ability to cover potential dynamic responses to herbivory.

The Type II and Mixed Saturating response were best-supported by Akaike Information Criterion (AICc) weights with the Type II being the favorite (Fig. 1). For the Mixed Saturating model, the estimated value of the exponent parameter $b$ is nearly 1, making these two models very similar in overall form across the range of herbivore damage. Additionally, when the intercept ($i$) is set as 1 across models (presuming no HIPL effect as a control), the AICc weights for the Type II and Mixed Saturating responses are approximately 0.85 and 0.14, respectively. Finally, additional analysis corroborates the support for the Type II response (see Methods section, Supplementary Notes 1 and 2). Therefore, the form describing HIPL that will be used in the full model will be $v(c, h) = \frac{1}{1+ch}$ (Fig. 2). Given the appreciable support for the Mixed Saturating form, we did analyze cases where $b > 1$. We also analyzed the effects of the other functional forms of HIPL. Analysis showed consistent results with those presented here (see Discussion section).

parameter which describes the intensity of the effect of $h$ on pollination.

Five potential models are considered and fit against the data: (1) Type I or linear decline response, (2) Type II declining response, (3) Type III declining response, (4) Mixed saturating decline, (5) Concave declining function (Fig. 1). Type I, II, and III functional responses are named as such due to their dynamic similarity to functional responses seen in predation and mutualistic interactions. The Mixed Saturating model tests the effect of a response model with a scalar multiplier, $c$, on $h$ and a potential non-integer exponent, $b$ (Fig. 1). The Concave function allows for the testing of a potential threshold effect. These response models were chosen based on their established use in the

**Full model description.** This model (Eq. 1) takes the form of three coupled ordinary differential equations with the following three variables: (1) population abundance of the shared flowering resource plant ($F$), (2) population abundance of the herbivorous insect ($H$), and (3) the population abundance of the insect pollinator of the flowering plant ($P$). Herbivory occurs through a typical Lotka–Voltera consumer–resource interaction with a Type II functional response, with the rate of herbivory labeled $r_H$. Though studies of the functional responses of herbivores have not focused on insects, a Type II functional response has been found in numerous taxa[31–33] and is a commonly assumed form used in many consumer–resource models[28]. Pollination of flowering plants by pollinators also incorporates a Type II functional response. This was first applied in Wright's 1989 modified model of stable mutualisms and has seen support from empirical studies[30,34,35]. Both the flowering plant and pollinator experience

**Table 1 Parameter definitions for Equation 1**

| Parameter | Definition |
|---|---|
| $\alpha$ | Strength of density dependence. Set to 0.1 |
| $r_F$ | Intrinsic reproductive rate of the flowering plant independent from pollinator $P$ |
| $r_H$ | Rate of herbivory of herbivore consuming flowering plant |
| $c_{FH}$ | Conversion rate of eaten plant biomass $F$ into herbivore $H$ |
| $h_H$ | Handling time of the herbivore on the flowering plant |
| $h_P$ | Handling time of the pollinator on the flowering plant |
| $b_F$ | Reproductive benefit of pollination visit for the flowering plant |
| $b_P$ | Reproductive benefit of pollination visit for the pollinator |
| $c$ | Degree of pollinator visitation reduction due to effect of herbivore |
| $d_F, d_H, d_P$ | Background death rates for flowering plant, herbivorous insect, the pollinator respectively |

Parameters are measured per individual per unit time. The 3 time dependent variables in the model are as follows: $F$-flowering plant population, $H$-herbivore population, $P$-pollinator population

**Table 2 All non-zero equilibria for Equation 1**

| Equilibrium | Description |
|---|---|
| 1) $F^* = \frac{r_F - d_F}{\alpha}, H^* = 0, P^* = 0$ | Only possible when $r_F > d_F$. F goes to carrying capacity. |
| 2) $F^* = \frac{d_H}{c_{FH}r_H - d_H h_H}, H^* = \frac{-c_{FH}(\alpha d_H - (d_F - r_F)(d_H h_H - c_{FH}r_H))}{(d_H h_H - c_{FH}r_H)^2}, P^* = 0$ | Only possible when $r_F > d_F$. P is eliminated from the system and F,H community persist in a steady state as a consumer–resource system. |
| 3) $F^* > 0, H^* = 0, P^* > 0$ | H eliminated from the system. F and P persist in a steady state. Parametric expression too large to write here. |
| 4) $F^* > 0, H^* > 0, P^* > 0$ | All 3 variables persist in a steady state. Parametric expression too large to write here. See analysis in Results section. |

Note the system is also stable at the 0-equilibrium, $F^* = 0, H^* = 0, P^* = 0$. All variable and parameter descriptions are given in Table 1

density-dependent growth as the populations are limited by space or nesting availability, respectively. The limitations on population growth due to density dependence scale with the parameter $\alpha$.

The flowering plant and pollinator receive a reproductive benefit of $b_F$ and $b_P$, respectively from pollination, which represents the conversion efficiency of the pollination visits. Baseline visitation rates, i.e., interaction rates between $F$ and $P$ when there is no herbivory, have a default value of 1 functioning as the visitation control value. Given that the $y$-intercept of the best fit form of $v(c, h)$ is approximately 0.96 (Fig. 1), this assumption is reasonable. Any small changes to this value would not qualitatively change the results presented below. To incorporate the functional form of HIPL, we assume that levels of herbivory are proportional to the density of herbivores. Pollination rates are therefore affected by, $v(c,H)$ where $H$ replaces $h$ and $c$ represents pollinators' sensitivity to herbivore presence/damage. Additionally, because $v(c,H)$ now considers herbivore density and not the resulting percentage of leaves damaged by herbivores, tested values of parameter $c$ will be higher than estimates in Fig. 1.

The flowering plant has an average rate of reproduction, independent of the focal insect pollinator population ($P$), represented as $r_F$. When $r_F$ is set to 0, the flowering plant is dependent on pollination from $P$ for fertilization so its reproduction rate is regulated by the parameter $b_F > 0$. This represents an obligate relationship with the mutualists in the model. Obligate mutualisms, while not common, are well documented[36–40] and serve as a foundation to understanding more complicated mutualistic networks in this context. Similar to obligate mutualisms, but more common[41–44], are specialized pollination mutualisms where some generalization exists, but the majority of visits with successful pollen deposition and fertilization is made up of a particular pollinator–plant pair. With some small positive value $\epsilon$, such that $r_F = \epsilon$, we can model a highly specialized pollinator mutualism, where $r_F$ contributes slightly to

plant reproduction and the pollinator is still dependent upon $F$. When $r_F > 0$ by a substantial amount ($r_F > \epsilon$), the flowering plant is able to produce some average amount of viable seeds through animal pollination unaffected by insect directed HI-VOC release (e.g., bat or bird), vegetative reproduction, or self-fertilization. Self-fertilization can be common in specialized pollination systems[45,46]. This condition models a generalist/facultative mutualism for the flowering plant. Pollination specialization can often be asymmetric[47–51], so in this model, only the flowering plant population's growth is allowed options outside the focal pollination mutualism with $P$. The full model is given in Eq. 1. All model parameters are listed and described in Table 1.

$$\frac{dF}{dt} = F\left(r_F + b_F v(c, H)\frac{P}{1 + h_P F} - \alpha F\right) - \frac{r_H F H}{1 + h_H F} - d_F F$$

$$\frac{dH}{dt} = \frac{c_{FH} r_H F H}{1 + h_H F} - d_H H$$

$$\frac{dP}{dt} = P\left(b_P v(c, H)\frac{F}{1 + h_P F} - \alpha_P P\right) - d_P P \tag{1}$$

$$v(c, H) = \frac{1}{1 + cH}$$

The model is formulated with a specialist herbivore population that doesn't gain any metabolic energy from any other plant species. While this limits the model's application to generalist herbivore species, specialist insect herbivores are very common[51,52]. Finally, while the model and form of $v(c,H)$ allow for a variety of mechanisms for HIPL, including HI-VOCS, this model does not include negative effects of HI-VOCS on the herbivore population, such as herbivore repellence and third trophic level interactions[53–55]. We argue this is acceptable, at least initially, as specialist herbivores often exhibit resistance to chemical repellence[56–58] and temporary herbivore repellence or control

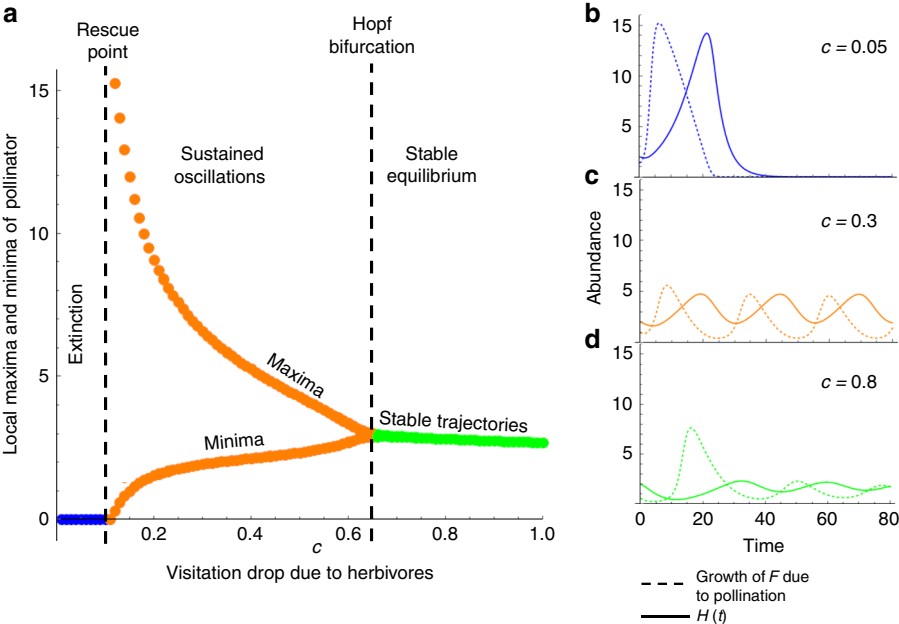

**Fig. 3** Effects of HIPL on community dynamics within an obligate mutualism. Changing community dynamics with varying degrees of pollinator visitation reduction due to herbivory (different values of parameter $c$). **a** A bifurcation diagram of the pollinator variable $P$ across values of $c$. Values which lead to extinction, sustained oscillations, and dampened oscillations are labeled and shown in blue, orange, and green respectively. The "rescue point" and Hopf bifurcation are marked with dashed lines. **b** Time series curves of growth of flowering plant population $F$ due to pollination $Fb_F v(c, H) \frac{P}{1+h_P F}$, dashed lines) and subsequent saturation of system with herbivores $H$ (solid lines) leading to extinction. **c** HIPL reduced growth of $F$ (orange dashed) and subsequently attenuated growth of $H$ (orange solid). **d** Highest level of visitation reduction leading to dampened oscillations and stable equilibria $F$ growth is shown in the green dashed line while the $H$ is shown in the solid green line. $d_F = 0.2$; $d_H = 0.28$; $d_P = 0.2$; $c_{FH} = 1$; $b_F = 1.565$; $b_P = 1.865$; $r_F = 0$; $r_H = 0.445$; $h_F = 1$; $h_P = 1$; $\alpha = 0.1$, all initial conditions = 2

would only reinforce the stabilizing effects of HIPL discussed in the Results sections below. Here we focus on the plant-mediated effect of herbivory on pollinator behavior and the resulting broader community dynamics.

**Model equilibria and pollination without HIPL.** There are four general equilibria for the 3 species of the model (Table 2). The equilibrium values for the three variables $F, H$, and $P$ are labeled $F^*$, $H^*$ and $P^*$ respectively across all equilibria. Equilibria 1, 2, and 3 are equilibria that have been studied in well-established work and will not be of focus here. Equilibrium 4 is the lone equilibrium in which all three variables can persist in a positive-valued steady state. The parametric expression of $F^*$ in Equilibrium 4 is $\frac{d_H}{c_{FH} r_H - d_H h_H}$. Expressions for $H^*$ and $P^*$ change depending upon the inclusion or exclusion of $v(c, H)$ and the status of the mutualism (see Supplementary Notes 3 and 4). The community can also go extinct such that all three populations in the system tend to 0. We will refer to this as the 0-equilibrium, representing full community extinction. Therefore, the two equilibria of interest are the 0-equilibrium and Equilibrium 4. Equilibrium 4 and persistent periodic oscillations for all 3 populations (stable limit cycles) will be referred to as "non-zero attractors." A non-zero attractor is any stable dynamic through time, which attracts nearby trajectories to it and results in the persistence of all populations. The full effect of HIPL and $v(c, H)$ on system persistence is made clear by first setting $c = 0$, making $v(c, H) = 1$ in Equation 1. This effectively eliminates the mechanism of HIPL from the model and verifies that previously described patterns[23–27] are reproducible with our model. More specifically, it shows that obligate and highly specialized plant–pollinator mutualisms can be destabilized and driven to extinction in their more basic theoretical formulation without trait-mediated indirect effects, such as HIPL (Supplementary Note 3).

**Obligate and specialized mutualisms with HIPL.** Keeping $r_F = 0$ (obligate mutualism), but setting $c > 0$ and including the effect of the Type II functional form of $v(c, H)$, greatly alters the obligate mutualism's response to higher rates of herbivory. Most notably, the mutualism is either unaffected or more resilient to comparatively much higher values of $r_H$ at the population level. In other words, the mutualism and the system overall can persist through much higher rates of herbivory. In fact, HIPL often creates a non-zero attractor where none existed before and consequently allows the mutualism to survive in systems that led to extinction when $c = 0$.

Using a bifurcation diagram with $c$ as the bifurcation parameter, we can see that low values of $c$ result in system extinction (Fig. 3a, b). Increasing the value of $c$, the system reaches the "rescue point," taking population trajectories from extinction to sustained oscillations (limit cycles) (Fig. 3a, c). The exact value of $c$, which becomes the rescue point depends on other parameters in the model and increases with higher rates of herbivory (Fig. 4). Yet, higher values of $c$ push the system to a Hopf bifurcation which merges the maxima and minima of the oscillations to the same point leading to a locally stable Equilibrium 4 and steady state dynamics (Fig. 3a, d). Therefore, the model shows the potential for HIPL to allow for community persistence at higher rates of herbivory and stabilized systems despite further reducing interaction rates among mutualists.

The mechanism of system persistence is apparent by considering the form of $v(c, H)$. Given that $v(c, H) = \frac{1}{1+cH}$ when $c > 0$, $v(c, H)$ and $H$ will oscillate asynchronously through time. In other words, pollination rates will only reach maximum levels when herbivore densities are low (Supplementary Note 4 and Supplementary Fig. 5). We can see the results of this asynchronicity by plotting the growth in $F$ due to pollination and the subsequent effect on the herbivore population at different

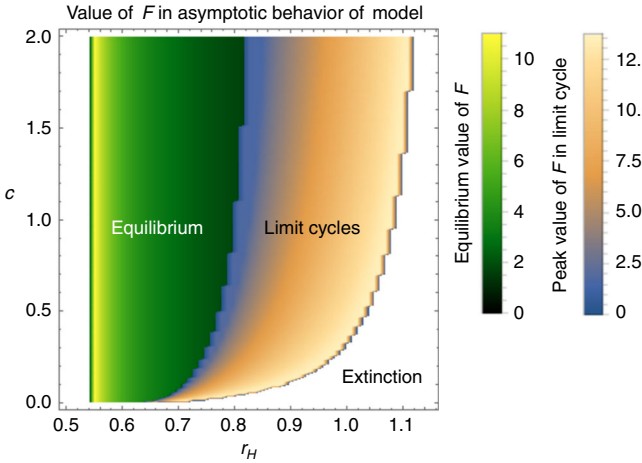

**Fig. 4** Effects of HIPL on community dynamics across parameter space. A two-dimensional bifurcation heatmap showing the abundance of $F$ (flowering plant) in the asymptotic behavior of the model shown as different colors across the $\{r_H, c\}$ parameter space in the obligate model. Where parameter combinations create stable equilibria, $F$ abundance is shown in the green color scale. Where values create stable limit cycles, $F$ abundance is shown in the sunset color scale. The switch between the two color schemes represents the Hopf bifurcation shown in Fig. 3. Values which lead to either extinction of $H$ ($r_H < 0.55$) or full system extinction (lower right portion of figure) are shown in white. $r_F = 0$; $b_F = 1.665$; $b_P = 1.695$; $c_{FH} = 1$; $d_F = 0.2$; $d_H = 0.5$; $d_P = 0.2$; $h_F = 1$; $h_P = 1$; $\alpha = 0.1$

levels of visitation reduction (Fig. 3b–d). When $c$ is below the rescue point (Fig. 3a, b), HIPL is weak and per-capita pollinator visitation rates are roughly steady despite high herbivore densities. Herbivory does not reduce pollination received by individual plants in this case. This causes a sharp increase in $F$ population growth, followed by a sharp rise in $H$ (Fig. 3b). This saturates the system with herbivores and the mutualism cannot recover. Therefore, despite the immediate benefit of the reproduction of individual plants, the subsequent increase in herbivores is substantial enough to eliminate the plant population and consequently the pollinator.

After the rescue point (Fig. 3a, c), pollinator visitation begins to decrease in response to higher $H$ loads. This reduces the initial growth of $F$ as plants receive less immediate pollinator visits (Fig. 3c). In turn, this reduces $\frac{dH}{dt}$ and the peak value of $H$ (Fig. 3c), keeping the population level of $H$ low enough for the system to persist in oscillations. Finally, past the Hopf bifurcation point, (Fig. 3a, d) pollinator visitation drops quickly even with moderate herbivory. This causes pollination-induced growth of $F$ to stay low and the $H$ population cannot continue to grow (Fig. 3d). The asynchronicity of $v(c,H)$ and $H$ creates a stabilizing effect, which rescues the system and induces sustained oscillations or stable equilibria depending on the level of pollinator aversion to herbivores/herbivory. In this case, the decrease in immediate plant reproduction is mitigated by the indirect control of the herbivore population. Limitation of pollinator visitation then actually has a net benefit to the plants and pollinators at the population and community level.

There is a clear expansion of the rate of herbivory (value of $r_H$), which the mutualism can withstand as the value of $c$ increases (Fig. 4, a two-dimensional bifurcation heat-map). In this particular formulation of the model, at the highest value of $c$ tested, the range of $r_H$, which the mutualism can withstand increases by ~4.33 times compared to the initial system, where $c = 0$ and $v(c,H) = 1$ (Fig. 4, see Supplementary Fig. 6 for $H$ and $P$). The highest sustainable value of $r_H$ reaches nearly double that of a system, where $c = 0$. Note, that the degree of this increase is also

affected by the values of other parameters (e.g., reproductive benefit of pollination to the mutualist populations).

Analogous rescue effects and community dynamics are producible in the highly specialized case where $r_H = \epsilon$ for some small positive value $\epsilon > 0$. Examples are available in Supplementary Note 5. Finally, while $c > 0$ can induce system persistence, it is not without some potential cost. Both rates of herbivory ($r_H$) and pollinator aversion to herbivory ($c$) can have significant effects on the volume of the basin of attraction of non-zero attractors. In other words, HIPL ($c > 0$) can create the potential for system rescue, but higher values of $c$ reduce the amount of initial system conditions which move toward non-zero attractors (Supplementary Note 6).

**Facultative mutualism with HIPL**. This section examines a system where the shared plant resource $F$ has a substantial non-zero growth rate independent from $P$ ($r_F > \epsilon > 0$) and there is visitation reduction ($c > 0$). This creates a system where the mutualism is obligate for the pollinator, but facultative/generalist for the flowering plant. In this case, while visitation reduction can still save the system from extinction, simulations show that the benefits of visitation reduction (especially for the pollinator, $P$) are dependent upon how much plant growth occurs independent from $P$ (i.e., the value of $r_F$).

When the value of $r_F$ is low and relatively small compared to the reproductive benefit of pollination, then HIPL can still indirectly control herbivore populations and rescue the system from extinction in a similar manner to the highly specialized case described above. However, when the system exists under a sufficiently high rate of herbivory ($r_H$) and a sufficiently high degree of visitation reduction ($c$) then higher values of $r_F$ can decrease pollinator abundance and push $P$ to a crash point. Again, these effects are displayed in a bifurcation diagram, this time across different values of $r_F$ (Fig. 5). For $0 \leq r_F \leq 0.73$ the system supports a stable pollinator population at Equilibrium 4 but with a monotonically decreasing abundance of $P^*$ as $r_F$ increases (Fig. 5a). While the idea that higher growth rates of one mutualist would limit its mutualistic partner seems unintuitive, the reason for this is the relationship between herbivore and pollinator populations when the system is stable at Equilibrium 4. Analysis shows that $P^* \sim \frac{1}{H^*}$ (Supplementary Note 4) due to herbivory reducing flower numbers and the effect of $v(c,H)$ (Fig. 5b). Also, $H^*$ was found to increase over this same range of $r_F$ (Fig. 5b). Moreover, in Equilibrium 4, $F^* = \frac{d_H}{c_{FH}r_H - d_H h_H}$, so while higher $r_F$ supports larger $H$ populations in equilibrium, there is no corresponding increase in the population of $F$. As the abundance of herbivores increases, so does the effect of HIPL. High effects of HIPL limit pollination interactions between $F$ and $P$ decrease the population growth of the pollinator. While this would cause both $P$ and $F$ to decline when $r_F$ is low, the high values of $r_F$ allow the plant–herbivore system to persist without the pollinator. In other words, the population level effects of HIPL on the plant and pollinator populations become decoupled in a more generalist/facultative mutualism. Therefore, increased intrinsic growth from the flowering plant can actually reduce pollinator abundance through the mechanism of HIPL. Sufficiently high values of $r_F$ increase $H^*$ to a level which pushes the pollinator population to extinction by pushing $P^*$ to 0 (Fig. 5a, b).

Further increases in $r_F$ induce limit cycles, as they would in a classic Lotka–Volterra system. As $F$ and $H$ oscillate, the amplitude of these oscillations can allow for windows of time where the pollinator population can grow. This occurs because higher $r_F$ creates larger, more dramatic oscillations in the plant–herbivore system. These large oscillations create higher peaks in $H$ but consequently result in lower minima values (bifurcation diagram

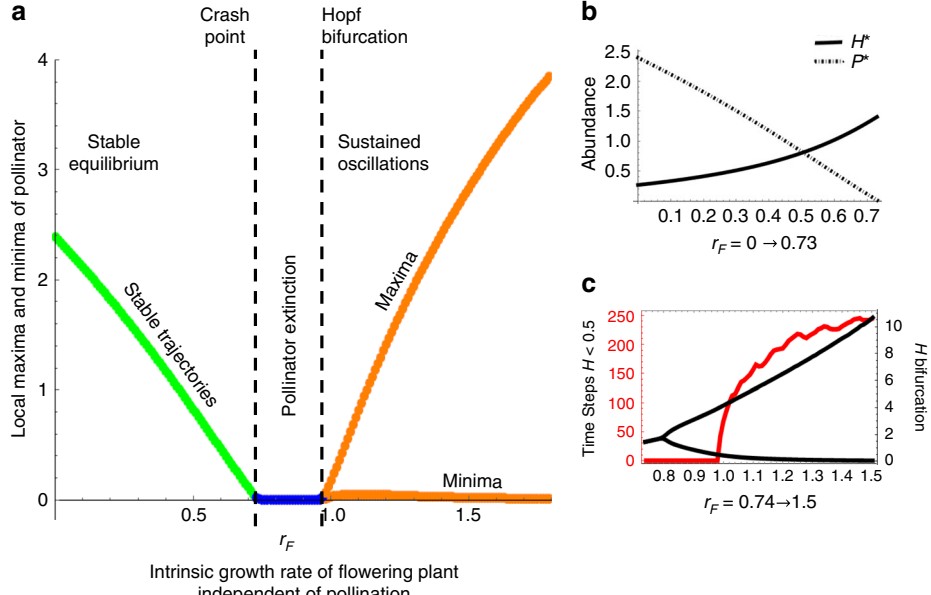

**Fig. 5** Effects of HIPL on community dynamics within a facultative mutualism. Changing community dynamics with increasing values of the parameter $r_F$ (intrinsic growth of $F$). **a** A bifurcation diagram for $P$ (pollinator) across values of the intrinsic growth rate of the flowering plant population ($r_F$). Values which lead to extinction, sustained oscillations, and dampened oscillations are shown in blue, orange, and green respectively. The "crash point" and Hopf bifurcation are marked with dashed lines. **b** Changes in $H^*$ (solid lines) and subsequent changes in $P^*$ (dashed lines) for $r_F = 0 \to 0.73$. **c** Black lines and right y-axis represent a bifurcation diagram of $H$. As $r_F$ increases past 0.8, the system and $H$ populations begin to oscillate with increasing amplitude. Red line and the left y-axis show the increase in the amount of time per simulation that $H < 0.5$. Results shown for $r_F = 0.74 \to 1.5$. The value of $r_F$ which marks longer times with lower $H$ abundance ($r_F \approx 1$) correspond to the value of $r_F$ which facilitates higher $P$ maxima shown in **a**. Other parameter values: $c = 1.44$; $r_H = 0.7$; $c_{FH} = 1$; $d_F = 0.2$; $d_H = 0.5$; $d_P = 0.2$; $h_F = 1$; $h_P = 1$; $b_F = 1.04$; $b_P = 0.85$; $\alpha = 0.1$

of $H$ in Fig. 5c). Lower minima values mean longer recovery times from low population abundances. This result in longer periods of time, where herbivore abundance is low in-between oscillatory population peaks. Heuristically, this can be shown by measuring the amount of time $H < 0.5$ as $r_F$ increases (Fig. 5c, red line). This increased time with low $H$ abundance increases the time $P$ can grow unencumbered by HIPL. This creates higher oscillatory peaks in $P$ abundance (Fig. 5a, Supplementary Note 7).

## Discussion

Understanding the functional responses of interacting organisms has been critical to the development and extension of theoretical foundations to empirical studies of herbivory[59], mutualisms[34,35], and especially predation[60,61]. However, the functional response of mutualist interactions (e.g., pollinators) across levels of antagonistic interactions (e.g., herbivory) has only recently become a research focus within the plant–herbivore interaction and community dynamics context. This increased interest largely rests on two conceptual pillars. First, interactions among members of a plant community are complex and removal or addition of players can have dramatic differential consequences[62]. Second, many of the interactions are mediated by plant metabolic responses to environmental stressors (e.g., herbivory) which broaden the arena in which plant-organismal interactions occur and further affects context dependency of functional links mediating interactions[1]. Measuring functional responses between ecological variables accounts for the fact that rates of interaction between species are not constant. In the case of HIPL, the functional response is particular in that the change in interaction rates between mutualists is mediated by a third party (the herbivore) interacting with the host plant. This is somewhat similar to trait-mediated indirect interactions, or TMII[63]. Here we show support for the hypothesis that pollinator visitation rates may decrease as a Type II function in response to herbivory. It had already been experimentally

verified that this effect can be primarily driven by a plant response to herbivory, HI-VOCs[18,22]. Understanding how pollination changes due to broader interactions within communities will be an important component in the study of pollination services[64].

When an antagonistic species limits the interaction rate and therefore the reproduction of individuals in a mutualistic species pair, it is not unreasonable to consider this a fitness loss for each of the mutualists. However, the model presented here suggests that indirect population and community effects in a flowering plant, herbivore, and pollinator community can present various challenges to this conclusion. HIPL is dynamic across time, increasing or decreasing in intensity with herbivore abundance. In numerous cases, HIPL limits population growth of both mutualists thereby temporarily and indirectly limiting herbivore abundance. This allows for the persistence of plant and pollinator populations despite the temporary decrease in individual fitness due to phenotypically plastic plant traits.

These indirect effects leading to persistent mutualist populations occur across much of the parameter space tested here, though some caveats should be noted. Even when pollination is reduced due to herbivory ($c > 0$), system persistence depends on relatively high values of reproductive benefit per pollinator visit for both the plant and/or the pollinator ($b_F$ and/or $b_P$). Also, sufficiently high herbivore attack rates ($r_H$) and/or low death rates ($d_H$) can still crash the system, though this can be counteracted by simply lowering herbivore conversion rates. Regardless of these limits in parameter space however, adequately high values of $c$ can expand the range of $r_H$ that the system can withstand (Fig. 4), creating non-zero attractors where none existed before. Sufficiently high $c$ can even lower the level of reproductive benefits of the mutualists ($b_F$ and $b_P$) required for community persistence (Supplementary Fig. 7). Mainly though, it is noteworthy that there is the potential for a decrease in mutualist interaction rates

and a subsequent decrease in population growth of one or both mutualists (plant and pollinator) to function as a mechanism for system persistence in the face of an antagonistic interaction. However, the results from the generalist/facultative model indicate that the effects of HIPL will become more complex when embedded into more complicated mutualistic networks.

Because herbivores and pollinators interacting with a particular plant species use the same information space[1] it was long hypothesized that plant traits are under conflicting selection to repel herbivores while still attracting pollinators[15]. The HIPL found in *S. peruvianum* was driven by HI-VOC-mediated information transfer between plants, herbivores and pollinators. The relatively strong negative ecological effect on plant fitness, poses the question why plants maintain such a strong induced, seemingly costly VOC emission in response to herbivory? Two principal hypotheses were suggested: (A) inducible volatile emission has additional functions in mediating interactions such as repelling herbivores, attracting natural enemies of herbivores or reducing plant damage through within plant signal transduction[18]. Alternatively (B) HIPL and the resulting reduced investment in seed production may be a mechanism for the plant to reduce opportunity costs potentially resulting from high seed production when herbivory limits resources. While this study does not specifically address these hypotheses, it offers an additional alternative hypothesis. We contend that plant-induced responses with ecological consequences like HIPL have broader indirect effects in a population or community context. Indirect effects can reduce the risk of extinction as well as the strength of natural selection against HI-VOC release because they limit reproductive ability of individual organisms such that population growth rates are maintained at sustainable levels in the community context, resulting in a net benefit for the individual interactors. Ecologically this has been hypothesized to be driven by two mechanisms. First, induced plant metabolic changes affect the carrying capacity of the system and so influence the system's potential for population cycles and outbreaks[65]. Second, chemical information transfer between organisms allows for behavioral responses in all interacting organism, which, in turn prevents populations from reaching critical densities[66].

Notably, the Type II form of $v(c,H)$ used in this model is not asserted to be the definitive functional form HIPL will take in nature. Other populations, species, or systems may react to herbivory in a Type I or Concave form. For example, the curve fitting analysis done with the data from Barber et al.[67] did not produce a single best fit functional response and may result in a different form with more data points (see Supplementary Note 1). The HIPL displayed in these data did not originate from HI-VOCs, but from direct physical effects of herbivory on flower attractiveness and mycorrhizal fungi colonization. Perhaps other mechanisms of visitation reduction may be prone to different functional forms. Additionally, the Barber et al. study system was a less specialized pollination system, and the two major pollinators were both well-known generalists (bumble bees and honey bees). This may also affect the functional form of HIPL and indicates there is a need to study these effects in more pollination mutualisms along the full degree of specialization and generalization.

Prompted by the possibility of other functional forms, we analyzed model dynamics using alternate functions for $v(c,H)$. Overall, these analyses show that other functional forms can consistently indirectly control herbivore population growth when used in Eq. 1 (Supplementary Notes 8–11). Only the Concave functional form was found to noticeably limit the range of community persistence in tested parameter space. This occurred because the Concave function leads to long delays in the reduction of pollination services until herbivores reach comparatively

high abundances, consequently, eliminating the indirect control of herbivore population growth. It would reasonable to assume that such a dynamic would also occur in the Mixed Saturating case when $b > 1$. While there is a similar delay in HIPL when $b > 1$, it's relatively limited and is followed by such a steep decline in pollinator visitation that the effective indirect control of herbivore populations can occur at lower values of $c$ as the value of $b$ increases (Supplementary Note 10). Moving forward, our results show that developing an understanding of the ecological consequences of metabolic changes in plants may require incorporating a fuller range of ecological complexity.

## Methods

**Study system.** The data used in this work comes from a series of field experiments on the Pacific slope of the Peruvian Andes conducted by Kessler et al.[18] using a wild tomato species, *Solanum peruvianum* (Supplementary Data 1). *S. peruvianum* is a self-incompatible species, which is attacked by a diverse set of herbivorous insects and pollinated by bees in the *Apidae*, *Colletidae*, and *Halictidae* families[68]. Bees on *Solanum* flowers, like those on other poricidal flowers, need to be behaviorally specialized because pollen, as the only pollinator reward, can only be harvested by the bees through vibratile ("buzz") pollination[69]. Herbivory of *S. peruvianum* was found to significantly lower pollinator visits through HI-VOC release. This limited pollination led to notable effects on plant fitness and was found to occur in response to actual herbivore damage or to pharmacologically induced VOC emission (application of methyl jasmonate in the absence of actual tissue damage)[18]. Although other traits can be important in mediating complex interactions, in this system HI-VOC emission fully explained the behavior of the bees and so the effects on plant fitness.

While a number of studies have found evidence of herbivory reducing the amount of pollination individual plants receive, these studies often use categorical treatments of pollination levels measured with and without herbivore damage[13,22,70]. Few have studied pollination across a continuous spectrum of herbivore damage as was done in Kessler et al.[18]. It is this approach that allows for the investigation into the functional form of HIPL across various levels of herbivory.

**Statistical analysis.** In order to ascertain the functional form of the negative correlation between herbivory and pollinator visitation, Kessler et al's data[18] has been broken into 11 sets. The first set (serving as the control) measures average pollinator visitation at 0% herbivore damage and is followed by 10 categories each grouped by taking the averages of herbivore damage and pollinator visitation in 10 percentage point steps (Fig. 2). This results in 11 averaged data points with standard errors on the $x$ (herbivore damage) and $y$ (pollination percentage) axes (Fig. 2). Given that the model used in this work is a spatially implicit meanfield model where parameters model average per-capita interactions across populations, the use of average effects is appropriate. Candidate models for the functional form of HIPL were fit to the data using nonlinear (weighted) least-squares estimates (nls) in the statistical software R and compared using Akaike Information Criterion (AICc) weights given their nonlinearity.

Analysis of this averaged/binned the data points presented in the Results section reveals that the Type II was the best fit candidate. This was verified by applying the same statistical analysis to the raw un-averaged data, where the Type II response was similarly found to be the best fit, though with lower AICc weights (Supplementary Note 1). Additionally, incorporating the standard error of the original averaged 11 data points into the nls regression and giving weights to each mean value also shows the Type II response to be the best fit (Supplementary Note 2).

It should be noted that the functional form of visitation reduction will likely differ across systems and communities. For example, a similar data set collected by Barber et al[67] put through the same analysis results in no conclusive support for any one response model over the others tested. See Supplementary Note 1 for a full account of the analysis on that data.

**Model background.** This relationship between herbivory and pollination was first considered in theoretical models by Jang[23]. However, in Jang's analysis, no explicit functional form was ever ascribed to this relationship. It was kept as a formless term for mathematical analysis instead of taking a Type II or Type III form for simulation. Furthermore, the dynamics-based analysis was specifically focused on the number of possible equilibrium points, the stability of those points, and the qualitative categories of possible dynamics (such as dampened oscillations or sustained oscillations). Given this basis and the goals of the analysis, Jang concluded that herbivore-induced reductions in pollinator visitation rates have no effect on the "qualitative" behavior of the model. In other words, HIPL would not change the number of equilibria or the types of dynamics that the model can potentially exhibit. We do not dispute Jang's results, but instead show that important distinctions reside in the "quantitative" change in dynamics. Jang's conclusions may explain other researchers' decision to not include pollinator visitation reduction into their models[25–27,71]. Sánchez-Garduño et al.[24] did describe

the potential inclusion of a function akin to $v(c, h)$ but set it equal to 1. Sánchez-Garduño and Breña-Medina[72] did include a sigmoidal Type III functional response rate of pollinator visitation decrease $\left(\sim \frac{1}{1+ch^2}\right)$ for a brief numeric consideration of possible types of mutualism–herbivore community dynamics. As more studies find that herbivory can produce significant plant-mediated interactive effects with other organisms (e.g., pollinators) and that it can result in significant declines in plant fitness, we must begin to delve further into the effects of this prevalent ecological relationship.

**Mathematical analysis**. Analysis was done through Mathematica 10 using NDSolve with Explicit RungeKutta methods[73]. Large scale analysis was facilitated by University of Michigan's FLUX computing core.

**Data availability**. The empirical data supporting the findings of this study and used to create Figs. 1 and 2 are available within the paper's Supplementary Information files. All the other data are available from the corresponding author(s) upon request.

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

## Acknowledgements

Thank you to Nicholas Barber for cooperation with open access data. Thank you to Marcio Duarte Albasini Mourao and the University of Michigan's Center for Statistical Consultation and Research (CSCAR). This material is based upon work that is partially supported by the National Institute of Food and Agriculture, U.S. Department of Agriculture, Multistate Grant under NE-1501 awarded to the Kessler Lab.

## Author contributions

A.K. and P.G.: Jointly studied the best fit form for HIPL. P.G.: Developed and analyzed the model. P.G.: Wrote the first draft of manuscript and figures. P.G. and A.K.: Jointly revised figures and manuscript.

## Additional information

**Competing interests:** The authors declare no competing financial interests.

