## [Peer review file · Nature Communications]

Reviewers' comments:

Reviewer #1 (Remarks to the Author):

I really enjoyed reading this paper. I read it as a field ecologist who works on both mutualistic and antagonistic networks, so I can't comment on the quality of the modelling, but I can state unequivocally that this is a seriously interesting question the authors ask, and that they have an excellent field system to which to the model can be applied.

The paper is well written, and while I don't understand the minutiae of the model (my limitations though, not the writing) the question, system and outcome are very well presented.

Bringing together mutualistic and antagonistic networks is a relatively new field (which started c. 2011ish I think with a review in Ecology Letters and a handful of data papers). What is great to see here is empirical and theoretical approaches working hand in hand. A data derived model will have a wider appeal than a straight theoretical study too (though there is obviously a place for these).

The objectives the work are clear and readily understood. I'm assuming that the model is fit for purpose and I like the discussion of the results which clearly presents the caveats of the approach and points out the limitations (e.g. the work is a long way off understanding the effect of these indirect effects at the whole community level. That said, this MS provides much more than an incremental increase in our understanding and it provides a valid mechanism by which plants, herbivores and pollinators all persist together (which is what they do in the field in most cases!). I think that the work will have a wide readership of herbivore and pollination people, network people and anyone interested in community persistence. I only have two small points to raise on the MS itself:

1) Make it clear early on that the Tomato system is a field system, it was relatively late in the paper that I realised that it was a wild species in the field in Peru, rather than a domestic tomato in a greenhouse (ie an artificial system). This is a feather in the cap of the project, so make it clear from the start.

2) Personally I hate acronyms. They save space but they are for the writer's convenience not the reader's and each takes me a while to learn – especially similar and new ones like HI-VOC and HIPL. If the word account allows, use the full term as it'll make the readers life much more pleasant.

Reviewer #2 (Remarks to the Author):

In this paper, the authors investigate the consequences of herbivore induced pollinator limitation on the coexistence of a dynamical model of a four species module, composed of a plant pollinated by a pollinator and consumed by an herbivore. To do so, they used a published dataset to assess the shape of the functional response of HIPL, and then they test the effect of the strength of HIPL effect on the coexistence of the module, considering two cases:

They first study the case where both the pollinator and the plant are specialist/obligate mutualism. They then study the case where the pollinator is specialist/obligate mutualism and the plant generalist/facultative mutualism.

They found that the functional response of HIPL was of the type II in the investigated dataset, and that including HIPL into a dynamical model creates a new feasible equilibrium that allow community persistence for higher rates of herbivory. This rescue effect is stronger in the specialist case. These results nicely illustrate how interaction modifier can strongly impact the dynamic of interacting species.

Although I liked the approach and found the topic timely, it seems to me that some of the hypotheses of the model are not well supported and as a consequence, some analysis of the sensitivity of the results to these hypotheses would be needed to strengthen the results.

The authors state that the functional form of HIPL can change from one community to another (L199-202) and that the type II was found in one out of two dataset used. This questions a bit the sensitivity of the presented results to the functional form of the HIPL.

To justify the two studied cases, the authors argue that specialized pollination mutualism is common (L234-237) and that specialization is asymmetric in pollination networks (L244-246). Although I agree with these statements, I disagree that they justify the first study case as if both the plant and the pollinator are reciprocal specialist, there is no asymmetric specialization. I agree that such specialist-specialist pollination interaction exist but they are the exception according to the literature cited by the authors. This questions the strength of the HIPL effect in natural systems since the presented results indicate that the benefit of HIPL decreases as plants are more generalist/facultative. Studying a case were pollinator is generalist/facultative and the plant specialist/obligate might be more relevant.

L129: 130, what do you mean by specialist bee species? Is their any published work suggesting that?

Reviewer #3 (Remarks to the Author):

Dear editor,

I have read the manuscript "Herbivore-Induced Pollinator Limitation: Functional reductions in pollination services and their surprising potential in mutualist communities" with much interest. The authors develop a theoretical model including herbivores, pollinators and plants to study the complex interactions between antagonists (herbivores) and mutualists (pollinators). A previous model showed that the type of relationship that describes the effect of herbivores on the pollination rates does not change the qualitative behaviour of the model. The contribution of this study is that it shows that the parameter range under which community persistence is possible is much larger when considering that herbivory has a negative effect on pollination rates as specified through a type II functional response. The manuscript is well written. I like the type of analysis and the use of field data to feed into the model. The study of how mutualists and antagonists coexist in the domain of herbivore-plant-pollinator system is topical. My main concern is about the choice of the functional response that relates herbivory and pollination rates and how that affect the results.

The choice of the type II functional response is based on field data. (By the way, I find the naming of type I,II,III confusing since it is not related to the intake rate of the predator and the density of the prey and the function is declining instead of increasing). The authors present two datasets. In one dataset the statistical support for either a type I, or a type II, or type III or one of the other two other response types is not strong, rather all models get about equal support from the data. Likewise, for the first dataset, the support for a type II over other types is not very strong. Yet, in the abstract (line 32) and in the discussion (lines 394-395) the authors claim that the data strongly matched a strongly match a Type II functional-response curve.

1) The choice for the type II functional response is based on binned data. Binning the data seems to favour the type II functional response (compared to the raw data) over the other type of responses. However, to me it is not clear why the data is binned in the first place (as the raw data seems to be available)? If binned data was used, how did the authors include the standard errors around each average in their statistical model? And if this is not done, how does inclusion of the

standard errors affect their findings? For the raw data, what error distribution was used? Since the data represents counts, poisson distributed or negative binomial distributed errors are appropriate (but I get the impression normally distributed errors are used). I would like to see an analysis with poisson or NB errors on the raw data and the support for each of the functional responses.

2) Based on the type II response curve the authors investigate the conditions under which pollinators, herbivores and plants coexist. Since the justification of their choice for the type II is rather weak, I want to know how sensitive the main finding of this study (herbivore induced pollinator limitation promotes persistence) is to the choice of the response type. I would recommend that they take the general form $1/(1+ch^b)$ as parameter b can be included in the analysis. My guess would be that when $b > 2$ the range under which community persistence occurs will be smaller because pollinator rates do not decline very rapidly at low levels of herbivory. For this reason herbivore densities will decrease less rapidly which could drive pollinators to extinction.

3) Thus the statistical support for a type II response is weak and it is unclear to what extent the main result is affected by this choice. Yet, it seems to be the key selling point of the paper: functional reductions in pollination services and their surprising potential in mutualist communities. Please provide evidence that this finding is robust to the assumptions made.

Methods:

1. Is the model (Eq. 1) the same as is provided in Jang (2002 JMatBiol 44(2) 129-149) with the exception that $v(c,h)$ is specified to be a type II? If not, please indicate how it differs from this model. If another model was used as basis, please mention that one,
2. The authors do not provide units for the parameters and the state variables. This makes it very hard to check the model and interpret the parameters and the parameter values.
3. How were the equations solved and which integration routine was used?
4. Parametrisation. How were the parameter values chosen? From fig. B1 I understand the herbivores occur in highest abundances, followed by pollinators, and finally plants. This seems strange to me. Or are the abundances of the herbivores, pollinators and plants expressed in different units? How does the function $v(c,h)$ look like when applying to the herbivore abundances that are typically found in the model (for some representative values of c)? In other words, is the scaled slope comparable to the scaled slope that was found in the field?
5. The parameter value for cfh is missing in the figures in the main text.
6. How is it possible that below rh values of 0.55 populations go extinct? I would guess that at low values of rh herbivores would go extinct, but not F and P ?
7. Fig 5. b,c I have difficulties understanding those graphs.

Reviewers' comments:

Reviewer #1 (Remarks to the Author):

I really enjoyed reading this paper. I read it as a field ecologist who works on both mutualistic and antagonistic networks, so I can't comment on the quality of the modelling, but I can state unequivocally that this is a seriously interesting question the authors ask, and that they have an excellent field system to which to the model can be applied.

The paper is well written, and while I don't understand the minutiae of the model (my limitations though, not the writing) the question, system and outcome are very well presented.

Bringing together mutualistic and antagonistic networks is a relatively new field (which started c. 2011ish I think with a review in Ecology Letters and a handful of data papers). What is great to see here is empirical and theoretical approaches working hand in hand. A data derived model will have a wider appeal than a straight theoretical study too (though there is obviously a place for these).

The objectives the work are clear and readily understood. I'm assuming that the model is fit for purpose and I like the discussion of the results which clearly presents the caveats of the approach and points out the limitations (e.g. the work is a long way off understanding the effect of these indirect effects at the whole community level. That said, this MS provides much more than an incremental increase in our understanding and it provides a valid mechanism by which plants, herbivores and pollinators all persist together (which is what they do in the field in most cases!). I think that the work will have a wide readership of herbivore and pollination people, network people and anyone interested in community persistence. I only have two small points to raise on the MS itself:

1) Make it clear early on that the Tomato system is a field system, it was relatively late in the paper that I realised that it was a wild species in the field in Peru, rather than a domestic tomato in a greenhouse (ie an artificial system). This is a feather in the cap of the project, so make it clear from the start.

We have added the following to clarify this point:

L 75

“This negatively affected plant fitness via reduced pollen deposition when measured in the field^{18,19}.”
Our description of the system in the Study System section has the words “field experiment” in the first sentence, so I'm not sure there is much else to add there.

The only other place to add notification that this is a field system is in the abstract, but the abstract already has a full word count. We'd be happy to add this there if we were allowed to slightly expand the word count.

2) Personally I hate acronyms. They save space but they are for the writer's convenience not the reader's and each takes me a while to learn – especially similar and new ones like HI-VOC and HIPL. If the word account allows, use the full term as it'll make the readers life much more pleasant.

We agree that too many acronyms make a paper difficult to read outside of its intended field. However, a few points make the elimination of acronyms difficult. HI-VOC (herbivore-induced volatile organic compounds) is an established acronym used frequently in the field. VOC is used even more. Both HI-VOC and HIPL stand for somewhat lengthy terms (herbivore-induced volatile organic compounds and herbivore induced pollinator limitation). We think too much repetition of those full terms would be a bit too much wordage.

However, adding a few more instances of the full terms certainly wouldn't be an issue. I think the reviewer's point that the acronyms sound similar is a worthy consideration and writing the full terms out more towards the beginning of the paper could help reduce any confusion. Please see lines 76 and 100.

Reviewer #2 (Remarks to the Author):

In this paper, the authors investigate the consequences of herbivore induced pollinator limitation on the coexistence of a dynamical model of a four species module, composed of a plant pollinated by a pollinator and consumed by an herbivore. To do so, they used a published dataset to assess the shape of the functional response of HIPL, and then they test the effect of the strength of HIPL effect on the coexistence of the module, considering two cases:

They first study the case where both the pollinator and the plant are specialist/obligate mutualism. They then study the case where the pollinator is specialist/obligate mutualism and the plant generalist/facultative mutualism.

They found that the functional response of HIPL was of the type II in the investigated dataset, and that including HIPL into a dynamical model creates a new feasible equilibrium that allow community persistence for higher rates of herbivory. This rescue effect is stronger in the specialist case. These results nicely illustrate how interaction modifier can strongly impact the dynamic of interacting species.

Although I liked the approach and found the topic timely, it seems to me that some of the hypotheses of the model are not well supported and as a consequence, some analysis of the sensitivity of the results to these hypotheses would be needed to strengthen the results.

The authors state that the functional form of HIPL can change from one community to another (L199-202) and that the type II was found in one out of two dataset used. This questions a bit the sensitivity of the presented results to the functional form of the HIPL.

The reviewer raises an interesting question regarding the consistency of the results to the functional form of HIPL. While we found the most support for a Type II HIPL functional response, the 'rescue effect' is possible with every type of HIPL. We can show this by producing similar results and figures as those presented in the main paper (e.g. the bifurcation heatmap in Figure 3) using the model with different functional forms for HIPL.

SHOWING THE CONSISTENCY OF THE RESCUE EFFECT OF HIPL ACROSS DIFFERENT FUNCTIONAL RESPONSES:

TYPE I FUNCTIONAL FORM OF HIPL:

Here is a similar figure to the main paper's Figure 3, but here we use the Type I functional response for HIPL and instead of the Type II functional response shown in the main paper. Therefore,

$$v(c, H) = 1 - c * H$$

As in the main paper, the interaction rate of pollinators and flowering plants is assumed to be 1 when herbivore abundance and damage is zero. A linear decline in pollinator visitation can be realized in the model by utilizing a Piecewise function in the differential equation such that:

$$v(c, H) = 1 - c * H, \text{ when } 1 > c * H$$

$$v(c, H) = 0, \text{ when } 1 < c * H$$

This piecewise formulation stops $v(c, H)$ from becoming negative at any time. Using the Piecewise function means that $v(c, H)$ decreases linearly with increased herbivore populations until it reaches 0 and stays there for any herbivore population that is greater than $1/c$. With this instantiation of the model, we can produce similar graphs as those presented in the main paper (i.e. Figure 3), showing that the rescue effect of HIPL is robust to the selection of a Type I functional response. These results are shown below in Review Figure 1.

Value of F in Asymptotic Behavior of Type I HIPL Model

Review Figure 1: A two-dimensional bifurcation heatmap showing the abundance of F (flowering plant) in the asymptotic behavior of the model using a TYPE I functional response for HIPL. $r_F = 0$; $b_F = 1.265$; $b_P = 1.4$; $c_{FH} = 0.7$; $d_F = 0.2$; $d_H = 0.25$; $d_P = 0.2$; $h_F = h_P = 1.1$; $a = 0.1$.

TYPE III FUNCTIONAL FORM OF HIPL:

Continuing to test the other functional responses, here we use the Type III functional response for HIPL and instead of the Type II functional response shown in the main paper. Therefore, $v(c, H) = \frac{1}{1+cH^2}$. As in the main paper, the interaction rate of pollinators and flowering plants is assumed to be 1 when herbivore abundance and damage is zero. Again we present the results in the bifurcation heatmap figure (similar to Fig 3 in the main paper). The Type III functional response can allow for the rescue effect over similarly large subset of the parameter space (Review Figure 2).

Value of F in Asymptotic Behavior of Type III HIPL Model

Review Figure 2: A two-dimensional bifurcation heatmap showing the abundance of F (flowering plant) in the asymptotic behavior of the model using a TYPE III functional response for HIPL. $r_F = 0$; $b_F = 1.465$; $b_P = 1.615$; $c_{FH} = 0.7$; $d_F = 0.2$; $d_H = 0.25$; $d_P = 0.2$; $h_F = h_P = 1.1$; $a = 0.1$.

MIXED SATURATING CASE:

The Type III functional response is a subset/case, of the Mixed Saturating Case. From a modeling perspective, there are 3 parameters to test in this case ($\{r_H, c, b\}$). Regardless, the Mixed Saturating case is readily able to produce the main rescue effect result described in the main paper, but the details are more involved. The Type I, II, III functional responses only had one parameter per function (c), so it was possible to make the 2-D bifurcation heatmaps. In this case, there is more than one parameter for the mixed saturating functional response (parameter c and parameter b):

$$v(c, b, H) = \frac{1}{1 + cH^b}$$

Therefore, the previous $\{r_H, c\}$ bifurcation heatmaps do not tell the full story and we will need to show multiple figures to detail the full dynamics. Initially we can see the effects of different values of c and b for set values of r_H in the model using a similar heatmap/bifurcation figure but with the axes representing c and b instead of $\{r_H, c\}$. We chose this as our initial step partially to address Reviewer 3's comments about $b > 1$ reducing the range of the rescue effect in the parameter space.

We test the effects of higher values in b such that $1 \leq b \leq 3$. Such a parameter sweep would encompass the Type III functional response, but allows us to make a full range of comparisons. First, we test a relatively low interacting version of the system where $b_F = b_P = 0.78$ and $r_H = 0.67$ (Review Figure 3). The results presented in Review Fig 3 show that higher values of b do not require higher values of c to support the persistence of the system (i.e. there is no reduction in the range of the HIPL rescue

effect with higher b). In fact, it seems, lower values of c may support persistent systems with higher values of b .

Review Figure 3: A two-dimensional bifurcation heatmap showing the abundance of F (flowering plant) in the asymptotic behavior of the model using the Mixed Saturating functional response for HIPL across values of b and c . Other parameter values: $b_F = b_P = 0.78$, $r_H = 0.67$, $c_{FH} = 0.7$; $d_F = 0.2$; $d_H = 0.25$; $d_P = 0.2$; $h_F = h_P = 1.1$; $a = 0.1$.

We then made a similar graph with higher interaction rates (including herbivory), to see if that is the case. In Review Figure 4 we see that the system persists through limit cycle oscillations at lower values of c with higher values of b . In other words, it shows that higher levels of b can actually allow for the rescue effect and system persistence with *lower* levels of c than would be required when $c = 1$ in the Type II functional response. The only caveat being that $b > 1$ does not seem to produce as much stable equilibria in this case.

Review Figure 4: A two-dimensional bifurcation heatmap showing the abundance of F (flowering plant) in the asymptotic behavior of the model using the Mixed Saturating functional response for HIPL across values of b and c . Other parameter values: $b_F = 1.45$, $b_P = 1.55$, $r_H = 0.91$, $c_{FH} = 0.7$; $d_F = 0.2$; $d_H = 0.25$; $d_P = 0.2$; $h_F = h_P = 1.1$; $a = 0.1$.

Once our university computing core finally came back online in a functional manner, we expanded this parameter sweep to include r_H so that we could see where persistence occurred across $\{c, b, r_H\}$ parameter space. This was a large parameter sweep involving simulations over 453,000 parameter combinations. The results of this parameter sweep showed that as the value of b increased, the area of the remaining $\{c, r_H\}$ parameter space where the community can persist, increases. This, along with Review Figure 4 indicates that $b > 1$ can lead to community persistence at lower values of c (please see the response to Reviewer 3's comments for further on this point).

CONCAVE FUNCTIONAL RESPONSE:

It is possible to recreate the rescue effect with the Concave Functional Response (Review Figure 5). However, the Concave functional response generally created the smallest parameter space in which the rescue effect could be found. This makes sense to given the long delay in reducing pollination. The concave model is the least supported direct curve fit we attempted, so we claim that the only functional response type that noticeable reduces the range of the rescue effect in the model does not seem well supported. With this we can claim that the main results presented in the paper are robust to most functional responses types.

Review Figure 5: $c = .13, .338, 0.868$. $rh = .58, bf = 1.095, bp = 1.095, cfh = 0.7, h = 1.1, q = 3$. The green line, orange line, and black line represent the flowering plant, pollinator, and herbivore respectively. The pink line is the value of the $v(c, H)$ function as a response to the herbivore population.

Summary:

As Reviewer 2 pointed out, these are important components to consider so we have added a substantial amount to the Supplementary Information. The newly labeled Supplementary Information C details the above results and more in about 10 manuscript pages of analysis. We feel the Supplementary Information is the most appropriate location for this analysis as direct inclusion into the main paper would hinder the flow of the major ideas presented and create some redundancy as similar results are rehashed with different functional responses. However, by including the detailed analysis of the other functional responses in Supplementary Information C, the pertinent information is available for the interested reader. Please this new section pasted below as part of the response to Reviewer 3's comments or in the Supplementary Info added to the submission.

We inform the reader of Supplementary Info C on **Line 458:**

“However, additional analysis indicates that the other functional forms of HIPL can consistently indirectly control herbivore population growth when used in Equation 1 (Supplementary Info C).”

To justify the two studied cases, the authors argue that specialized pollination mutualism is common (L234-237) and that specialization is asymmetric in pollination networks (L244-246). Although I agree with these statements, I disagree that they justify the first study case as if both the plant and the pollinator are reciprocal specialist, there is no asymmetric specialization. I agree that such specialist-specialist pollination interaction exist but they are the exception according to the literature cited by the authors. This questions the strength of the HIPL effect in natural systems since the presented results indicate that the benefit of HIPL decreases as plants are more generalist/facultative. Studying a case were pollinator is generalist/facultative and the plant specialist/obligate might be more relevant.

On line 236 we state:

“Obligate mutualisms, while not common, are well documented⁴⁰⁻⁴⁴ and serve as a foundation to understanding more complicated mutualistic networks in this context.”

So we do state that there is a comparatively smaller amount of completely obligate pollination mutualisms than generalists, or species that show asymmetric specialization. The next line states that specialization (not necessarily, entirely obligate) is more common than completely obligate pollination. That is why the epsilon (ϵ) is used in the model analysis described in the Results section and detailed in the Supplementary Information B. This was incorporated to allow for a degree of asymmetric generalization between the flowering plant population and pollinator populations. We took this one-step further by then analyzing the facultative pollination model ($r_F > \epsilon > 0$).

Part of point looking at specialist-specialist relationships is they are theoretically found to be prone to disturbance and instability (as described in the main paper's Intro). So, while they are less common than generalists, it is worth studying the mechanisms of their persistence. It also reveals the interesting dynamic of the rescue effect in HIPL. Establishing this principle and studying its intricacies in a more symmetrically specialized system is needed to scaffold future work as a first step when expanding these studies to larger, more generalist/facultative systems. While we certainly agree that it would be

worthwhile and interesting to study the effects of HIPL further in various more expansive and generalized systems, it is beyond the purview of this study. It would be introducing additional complexities in a paper were we already introduce a novel finding.

However, as we agree that there is merit in these further studies, we have added the following the Discussion section:

Line 464:

“The Barber et al study system was a less specialized pollination system, and the two major pollinators were both well-known generalists (bumble bees and honey bees). This may affect the functional form of HIPL and indicates there is a need to study these effects in more pollination mutualisms along the full degree of specialization and generalization.”

Also, we do not assert that HIPL necessarily becomes “weaker,” only that the effects are more complex. Either way, HIPL changes the dynamics of the system. The overall qualitative dynamics of the system are different with and without HIPL.

L129:130, what do you mean by specialist bee species? Is their any published work suggesting that?

Thank you for this comment. Our wording was not entirely clear. In order to appropriately describe the study system that was used in the model, we chose to elaborate on the specific pollination system, but did not do this in enough clarity. The “specialization” here refers to the fact the suitable pollinators of tomato flowers need to be behaviorally specialized. Tomatoes as all *Solanum* flowers have poricidal anther cones that require bees to vibrate their wings and with them the anther cones to harvest pollen, which is the only pollinator reward. By far not all bee species can perform this vibratile or “buzz” pollination and requires some level of behavioral specialization. We clarified that point by altering the text stating on Line 129 to:

“*Solanum peruvianum* is a self-incompatible species which is attacked by a diverse set of herbivorous insects and pollinated by bees in the Apidae, Colletidae, and Halictidae families²⁸. Bees on *Solanum* flowers, like those on other poricidal flowers, need to be behaviorally specialized, because pollen, as the only pollinator reward, can only be harvested by the bees through vibratile (“buzz”) pollination²⁹.”

We included some explanatory citations:

28) Chetelat R.T., Pertuze, R.A. Faundez, L., Graham, E.B., Jones, C.M. Distributions, ecology and reproductive biology of wild tomatoes and related nightshades from the Atacama Desrt region of northern Chile. *Euphytica* 167, 77-93. DOI 10.1007/s10681-008-9863-6. (2009).

This publication describes some of the specialized bee species as well as the relatively high degree of pollinator specialization in *S. peruvianum* even compared to other species of *Solanum*. This stems from the particular shape of the anthers requiring the vigorous vibrating of flowers to release pollen, something only few bees are equipped to do.

29) De Luca, P. A., & Vallejo-Marin, M. What’s the 'buzz' about? The ecology and evolutionary significance of buzz-pollination. *Current Opinion in Plant Biology*, 16(4), 429–435. <https://doi.org/10.1016/j.pbi.2013.05.002> (2013).

More background information on the specialized behavior of buzz-pollination.

Reviewer #3 (Remarks to the Author):

Dear editor,

I have read the manuscript “Herbivore-Induced Pollinator Limitation: Functional reductions in pollination services and their surprising potential in mutualist communities” with much interest. The authors develop a theoretical model including herbivores, pollinators and plants to study the complex interactions between antagonists (herbivores) and mutualists (pollinators). A previous model showed that the type of

relationship that describes the effect of herbivores on the pollination rates does not change the qualitative behaviour of the model. The contribution of this study is that it shows that the parameter range under which community persistence is possible is much larger when considering that herbivory has a negative effect on pollination rates as specified through a type II functional response. The manuscript is well written. I like the type of analysis and the use of field data to feed into the model. The study of how mutualists and antagonists coexist in the domain of herbivore-plant-pollinator system is topical. My main concern is about the choice of the functional response that relates herbivory and pollination rates and how that affect the results.

The choice of the type II functional response is based on field data. (By the way, I find the naming of type I,II,III confusing since it is not related to the intake rate of the predator and the density of the prey and the function is declining instead of increasing).

We named them this way due to mathematical formulation of their functional forms being so similar to original Type I, II, and III's forms in consumption interactions. We are not against changing this, but we do think the above reasoning makes sense. The chosen names describe the manner in which these functions change monotonically, similar to how they are used in predation.

However, if the Reviewers/Editors are in agreement, changing the names is something we can certainly do. Here is a list of possible suggested changes:

- 1) Linear
- 2) Exponential Decay
- 3) Delayed Decay
- 4) Mixed Delay
- 5) Threshold (or just keep Concave)

The authors present two datasets. In one dataset the statistical support for either a type I, or a type II, or type III or one of the other two other response types is not strong, rather all models get about equal support from the data. Likewise, for the first dataset, the support for a type II over other types is not very strong. Yet, in the abstract (line 32) and in the discussion (lines 394-395) the authors claim that the data strongly matched a strongly match a Type II functional-response curve.

We should have been clearer here and mentioned that we were referring to the analysis using the binned presented in the main paper. While we address much of this concern below, we also made changes to main paper in order to better match the statistical results with our word choices.

For example:

Line 32: Changed strongly to best..

“We show these herbivory-induced decreases in pollination to individual plants best match a Type II functional-response curve.”

Line 402: Removed the word strong, added the word may:

“Here we show support for the hypothesis that pollinator visitation rates may decrease as a Type II function in response to herbivory.”

Line 194: Removed the word clear:

“The Type II and Mixed Saturating response were the best supported by AICc weights with the Type II being the clear favorite (Table 1).”

1) The choice for the type II functional response is based on binned data. Binning the data seems to favour the type II functional response (compared to the raw data) over the other type of responses. However, to me it is not clear why the data is binned in the first place (as the raw data seems to be available)? If binned data was used, how did the authors include the standard errors around each average in their statistical model? And if this is not done, how does inclusion of the standard errors affect their findings? For the raw data, what error distribution was used? Since the data represents counts, poisson distributed or negative binomial distributed errors are appropriate (but I get the impression normally distributed errors are used). I would like to see an analysis with poisson or NB errors on the raw data and the support for each of the functional responses.

There are a few points raised in this paragraph. We have tried to separate and organize our responses here.

Binning and average rates of interaction:

We addressed the binning of the data in the original paper at what was line 178:

“Given that the model used in this work is a spatially implicit meanfield model where interactions are averaged across the population, the use of average effects is appropriate.”

Well-mixed meanfield models with coupled differential equations are a common and well-established ecological modeling tool. They date back to Lotka-Volterra models and see continued use today in ecology and epidemiology.

These models use the per-capita interaction rates between populations to look at the fundamental dynamics of these systems, without complicating stochasticity or distributions of interaction rates/types. Per capita values represent *average* values for individuals in a population. For the visitation rate in the model, v , we are looking at a change in an average rate (visitation) across a gradient of herbivory. Therefore, we used bins to get an estimate of the average rate of visitation across the range of herbivory. Since, other interaction rates in all models of this type represent similar per capita/average rates, this creates a similar type of interaction between all populations across all interactions.

We have changed this passage to better connect these ideas:

Line 181:

“Given that the model used in this work is a spatially implicit meanfield model where parameters model per-capita interactions, the use of average effects is appropriate.”

Additionally, using the average rates and bins here is a better way to understand the expected value of pollinator visitation percentage at a level of herbivory. That’s not to say that stats on the full model doesn’t give expected values, but these are field measurements, not a laboratory experiment. So there are numerous other complicating factors that can interfere with each individual flower’s presence or absence of pollinator markings on flowers. It is possible that some flowers were attacked after being pollinated. Some flowers may be visited by bees that need immediate nectar after long foraging flights, despite some degree of herbivore damage. Other flowers may have been far away from pollinating bees and missed pollination despite having low herbivore damage. Averaging does reduce the influence of such instances in the curve fitting process. Finally, the binned data also presents a clear figure for the reader when printed out (main paper Figure 1). We argue this is useful in communicating the ideas of herbivores inducing the decline in pollinator visits.

Further Statistics on Raw Count Data:

While we do think there is merit in utilizing binned data, the Reviewer raises a good point about using the Poisson distribution on the raw count data. The mean and variance are similar because we are dealing with rates, given the normalization of each of the variables (by the number of flowers and the number of leaves for pollination and herbivory respectively). Therefore, we used the Poisson distribution.

Our original analysis on the binned data used the nls function in R which is a nonlinear fit with least squares. We used this same process on the raw data and presented this in the original analysis presented in the Supplementary Material. To further address the statistical analysis presented in the paper, we took our analysis to the University of Michigan’s Center for Statistical Consultation and Research (CSCAR, <http://cscar.research.umich.edu/>) to make certain we were addressing possible concerns.

As we were working on Reviewer 3’s comments, it became somewhat unclear as to what the Reviewer was asking for in this instance. The most common type of fit here would be to use a linear equation to estimate the dependent variable using the Poisson error distribution with the log link function. This is what our statistics consultants suggested and is what they interpreted the comments of the Reviewer to mean. However, we thought it possible that the reviewer may have been asking for a

generalized fit of the nonlinear functional forms using a Poisson distribution. While this second option did not seem as likely and raised concerns with our statistical consultant, we did both.

First, we used a glm in R to fit the raw count data of pollinated flowers offset by the number of available flowers against the degree of herbivore damage. Doing so produces the fit and summary stats shown in Review Table 1. The predicted fit is plotted below with ggplot (Review Figure 6). The shape of the predicted fit here does mimic the exponential decay relationship modeled by the Type II functional response. However, the AIC value is comparatively high and actually has the worse support than any of the other models tested.

Review Figure 6: Plotting the predicted relationship between herbivore damage and pollinator visited flowers derived from the generalized Poisson fit using the raw count data.

Fitted models to real data w/ intercepts	Estimated Parameters	Significance and Fit	AICc	AICc weight
1.)Linear: $\sim ch + i$	$c = -0.64,$ $i = 0.8583$	$p = (c)6.98e^{-13}$ $(i)2e^{-16}$ $R^2 = .2049$	168.3532	0.21497
2.)Type II: $\sim \frac{i}{1 + ch}$	$c = 1.7577,$ $i = 0.9440$	$p = (c)7.3e^{-7}$ $(i)2e^{-16}$	166.8308	0.46021
3.)Type III: $\sim \frac{i}{1 + ch^2}$	$c = 2.3514$ $i = 0.8282$	$p = (c)6.1e^{-6}$ $(i)2e^{-16}$	169.8738	0.10050
4.)Mixed Saturating: $\sim \frac{i}{1 + ch^b}$	$c = 1.8948$ $b = 1.1964$ $i = 0.9164$	$p = (c)1.68e^{-5}$ $(b).000163$ $(i)2e^{-16}$	168.3841	0.21167
5.)Concave: $\sim i * \left(1 - \frac{h}{100}\right)^c$	$c = 0.6062,$ $i = 0.8396$	$p = (c)4.52e^{-9}$ $(i)2e^{-16}$	174.0195	0.01265
6.) Poisson fit on count data:	Intercept= 0.9190 $\beta = -0.9983$	$p = (\text{Intercept})0.154$ $(\beta)1.43e^{-12}$	798.8534	2.6362 e^{-138}

Review Table 1: Summary statistics of model fits on raw data.

Second, to address the second possibility, we considered the fit of the non-linear functions using a Poisson error distribution. Our statistics consultant did not consider this to be a typically valid test given the addition of a non-linear function on top of the log link function that accompanies these fits. However, we did want to take care to address possible reviewer concerns, so we did run these model fits using the gnm package which can allow for generalized fits of nonlinear models using custom functions assigned in R.

When testing the Type II functional response, we assigned the function and ran the fit using the gnm package. The AIC value for the model rises to all the way to 805.48. This is noticeably higher than any of the previous tested models. Our statistical consultant was not in favor of this method given the mix of the two functional connections between the dependent and independent variables, and said that was likely, at least partially, responsible for the weaker fit.

Overall, when compared to the generalized Poisson fit, the Type II model is still the one with the strongest support when using the full count data set without bins. While this is not an official line of evidence, our statistical consultant did support the Type II case as the best-supported functional form through both the binned and non-binned data, for what it is worth. However, when compared to the bins, the reviewer does have a point in saying the support for the Type II does decline. When we originally said the data showed strong support for a Type II response in the original submission, we were referring to the binned fits. Therefore, there is merit in testing the other functional response types in the model. We did this and found the rescue effect of HIPL across large sections of parameter space, similar to the Type II functional response. Please see the response to Reviewer 2's comments and to your comments below.

Also, on the stats side of things, please note that we rescaled the binned analysis to range from 0.0 to 1.0 instead of 0 to 100 percent. It creates better consistency with the raw count data which is presented in that manner (0.0 to 1.0 range).

To include these analyses into the paper, we altered Supplementary Information A. Please see below:

///Changes made to Supplementary Information A

Supplementary Info A - Curve fitting on raw data:

In this work, two data sets were used from studies which investigated the effects of herbivory on pollinator visitation across a continuous spectrum of herbivory. The two studies used were Kessler et al 2011 & Barber et al 2012. While a number of studies have found evidence of herbivory reducing the amount of pollination individual plants receive, those studies often use categorical treatments of pollination measured with and without herbivore damage (Krupnik et al 1999; Adler & Irwin 2005; Kessler & Halitschke 2009). Few have studied pollination across on a continuous spectrum of herbivore damage as was done in these two studies. This data was used to curve fit and find the best possible support for the form of functional response of pollinator visitation rates to different levels of herbivory. As written in the main paper, the functional response is labeled $v(c, h)$ where h is the percentage of herbivore damaged leaves and c is parameter which describes the intensity of the effect of h .

Statistics on the raw data from each study can be found below. Each data was fitted against 6 models.: 1.) Type I or linear decline response, 2.) Type II declining response, 3.) Type III declining response, 4.) Mixed Saturating decline, 5.) Concave declining function, 6.) a generalized Poisson fit (see Table 1). Type I, II, and III functional responses are named as such due to their dynamic similarity to functional responses seen in predation and mutualistic interactions. The Mixed Saturating model tests the effect of a response model with a scalar multiplier on h and a potential non-integer exponent (see Supplementary Table A1). The concave function allows for the testing of a potential threshold effect. These response models were chosen based on their established use in the theoretical literature, their shown applicability in other interactions (such as predation and mutualist interaction), and their ability to cover potential dynamic responses to herbivory. Finally, the Poisson fit allows us to compare the functional response to a more traditional test of this type of count data. Models were fit to the data in the statistical software R and compared using AICc weights given their nonlinearity.

Supplementary Info A.1 - Kessler et al 2011:

Experiments in Kessler et al 2011 were conducted in Peru. This field study measured the proportion of flowers with pollination marks as a proxy for pollinator visitation and as a function of herbivore damage in the wild tomato *Solanum peruvianum*. Pollination marks were measured in relation

to herbivore leaf damage experienced by individual *S. peruvianum* plants. *S. peruvianum* is attacked by a diverse set of herbivorous insects and pollinated by specialist bees in the Apidae, Colletidae, and Halictidae families. For more information, please see the original paper (Kessler et al 2011).

The results of curve fitting the 6 candidate models to Kessler's raw data are displayed in Supplementary Table A1. This shows no entirely definitive support for a single model. The Linear, Type II, and Mixed Saturating models are all shown to have some comparable support. However, as with the results shown in the main paper, the Type II response has the highest support and the Mixed Saturating Model has a very similar form to the Type II ($b = 1.196$). We also note that the shape of the Poisson predicted fit does mimic the exponential decay relationship modeled by the Type II functional response. While the level of support in the raw data for the Type II form is more limited, the above reasoning and the results described in the main paper lead us to argue that the Type II response is the best suited functional response form from this data set. It should also be noted that not allowing the y-intercept to vary and fixing it to 100% increases the AICc weight of the Type II functional response to 0.85 and 0.62 in the averaged data fit and raw data fit respectively.

Supplementary Table A1: Model fitting to original data from *Solanum peruvianum* field experiments in Peru. Curve fittings of six candidate response models to Kessler et al 2011 data: Type I/Linear, Type II, Type III, Mixed Saturating, Concave, Poisson fit. Here h represents the level of herbivory. The parameters c and b determine the shape of the curve and i is the intercept. Estimated parameters that are significant have their p values bolded. The Type II functional response has the highest Akaike Information Criterion weight of 0.46021.

Fitted models to real data w/ intercepts	Estimated Parameters	Significance and Fit	AICc	AICc weight
1.)Linear: $\sim ch + i$	$c = -0.64,$ $i = 0.8583$	$p = (c)6.98e^{-13}$ $(i)2e^{-16}$ $R^2 = .2049$	168.3532	0.21497
2.)Type II: $\sim \frac{i}{1 + ch}$	$c = 1.7577,$ $i = 0.9440$	$p = (c)7.3e^{-7}$ $(i)2e^{-16}$	166.8308	0.46021
3.)Type III: $\sim \frac{i}{1 + ch^2}$	$c = 2.3514$ $i = 0.8282$	$p = (c)6.1e^{-6}$ $(i)2e^{-16}$	169.8738	0.10050
4.)Mixed Saturating: $\sim \frac{i}{1 + ch^b}$	$c = 1.8948$ $b = 1.1964$ $i = 0.9164$	$p = (c)1.68e^{-5}$ $(b).000163$ $(i)2e^{-16}$	168.3841	0.21167
5.)Concave: $\sim i * \left(1 - \frac{h}{100}\right)^c$	$c = 0.6062,$ $i = 0.8396$	$p = (c)4.52e^{-9}$ $(i)2e^{-16}$	174.0195	0.01265
6.) Poisson fit on count data:	Intercept= 0.9190 $\beta = -0.9983$	$p = (\text{Intercept})0.154$ $(\beta)1.43e^{-12}$	798.8534	2.6362 e^{-138}

Supplementary Figure A1: Plotting the predicted relationship between herbivore damage and pollinator visited flowers derived from the generalized Poisson fit using the raw count data.

///End of changes made to Supplementary Information A

The Poisson fit was also included for the Barber data set in the revised Supplementary Info A.

Incorporating Uncertainty:

The reviewer raises an interesting question regarding how our results might change regarding the incorporation of a range of interaction rates around an average rate for HIPL as inferred by the standard error around the model estimates shown in Figure 1 in the main paper.

There are multiple ways this could potentially be addressed; we detail two below:

**Note:* the simulations below use the TYPE II functional response, given the findings of the binned and raw data analysis.

First Method:

Each level of herbivory comes with a standard error around the mean visitation level, implying some range in the reduction of $v(c, H)$ per level of herbivore damage through HIPL. This range can be attained by allowing different levels of the parameter c . In other words, by having the value of c vary around some mean value according to a predetermined distribution for each time step will allow HIPL to occur within a range of values for each level of herbivore presence/herbivory. As an addition to our current model this would look like:

$$v(c, H) = \frac{1}{1 + cH}$$

$c \in \{Distribution\}$

c equals a number drawn from a user determined distribution at each time step with user determined parameters (e.g. mean) such that the value of c then varies around the chosen parameter value.

This distribution could be normal, uniform, etc. and all parameters in the chosen distribution can be controlled. The core dynamic of HIPL indirectly regulating herbivore growth leading to system persistence is readily apparent in this version of the model (see Review Figure 7 below):

Review Figure 7: Here we vary the value of c and by proxy $v(c, H)$ according to a normal distribution. The parameter value for c shown in each figure is the average value of c from the distribution. In the time series, green lines represent F , orange lines represent P , and black lines represent H .

We can also expand the range of the distribution to increase the variation around the mean value of c . Given the resultant higher variation in $v(c, H)$, higher values of c are then required to create smaller amplitude oscillations (see Review Figure 8 below).

Review Figure 8: Here we vary the value of c and by proxy $v(c, H)$ according to a uniform distribution within a larger set range. The parameter value for c shown in each figure is the middle value of the range of the uniform distribution. In the time series, green lines represent F , orange lines represent P , and black lines represent H .

Looking at Figure 1 in the original submission, we can see that the standard error around $v(c, H)$ slightly increases at higher levels of herbivory. In other words, the range of HIPL's effect on pollination would increase with higher amounts of herbivory. This can be incorporated into the model by allowing the range of the variation in c (and therefore $v(c, H)$) to scale with the abundance of the herbivores (H). In this case too, the rescue effect of HIPL is readily apparent (see Review Figure 9).

Review Figure 10: An example of the rescue effect while the range of variation in c scaled linearly with the abundance of herbivores (H). The parameter value for c shown in each figure is the middle value of the range of the uniform distribution. The range of the distribution scales linearly with the population abundance of the herbivore. The scaling of the range scales equally in the positive and negative direction, so the middle value of the range of c stays the same regardless of the herbivore abundance. It might be somewhat difficult to see here, but curves become the least smooth at the tops of the peaks in herbivore oscillations. In the time series, green lines represent F , orange lines represent P , and black lines represent H .

Second Method:

To *fully* incorporate uncertainty into a model would require integrating a range in rates across other interactions and ecological “actions” beyond just the pollinator visitation, and its reduction due to HIPL. That is, all ecological “actions” modeled (herbivory, birth, death) would need to vary such that each action in the model occurs within a range. Furthermore, these ecological “actions” would need to be allowed to vary on an *individual* level, or at least the model would need to mimic this level of detail.

We took steps to implement such a model through the use of stochastic equations modeled with the Gillespie algorithm. The Gillespie algorithm was originally used to model biochemical reactions that did not fit well to deterministic models. Instead, the Gillespie algorithm allows for a discrete simulation of a system where each single reaction is explicitly simulated. We can use this to our advantage by applying it to stochastic versions of the equations in our model. In this instantiation of the model then, rates function as probabilities where the average chance of occurrence can be represented as a population “rate,” but the individual level occurrence differs from action to action. That is, one herbivore may attack a flowering plant, but another may not. Or, given the interest in HIPL here, herbivory on a plant may deter one pollinator but not another. The actual occurrence of any of these interactions would depend on the probabilistic rate parameters given to the model, where higher rates mean higher chances of consistent occurrence.

After creating and running simulations with this model, the rescue effect of HIPL was readily apparent (see Review Figure 11). Given the variation in the occurrence of each interaction/action in this stochastic version, there is no true equilibrium. However, temporary reduction in pollination due to higher prevalence of herbivores does allow for system persistence. This comes through the similar indirect control of the herbivore population so that it does not reach high values. This is a similar dynamics as dampening oscillations. An example of this is visible in Review Figure 11 where the system fails when $c = 0$ and there is no effect of HIPL, but is able to persist when $c = 0.25$ and HIPL effects pollination rates.

Review Figure 11: An example of the HIPL rescue effect in the Gillespie algorithm version of the system. In the time series, green lines represent F , orange lines represent P , and black lines represent H .

Summary:

Analysis of either of these models would address interesting and important questions. However, while these questions are certainly related, they are also distinct from the dynamics oriented analysis in the original paper. Furthermore, a detailed explanation and complete analysis of either model would require a very large addition of content to the current manuscript. To do a complete analysis, these models would need to be run a large number of times to achieve a strong approximation of the master equation for each parameter set. With U of M's computing core currently experiencing problems, the time required for this would likely be extended. Either model, especially the Gillespie algorithm iteration, should merit a complete exploration in its own distinct manuscript/paper. In fact, such research is planned for the future.

For the reasons described above, we do not feel it would be suitable to add all the content to the paper. However, we did want to include explanation of these preliminary results here in our response to address the reviewer's comments, as they are worth considering. We feel that it may be appropriate to include a supplementary description of some of the above preliminary results from the first method since it builds off of the architecture and model techniques from the model presented in the paper. We argue the second method would most appropriately fit into a separate manuscript, but hope that the reviewers agree that this preliminary analysis supports the fact that this result is not just model dependent and can occur with the inclusion of uncertainty and range around the parameters.

2) Based on the type II response curve the authors investigate the conditions under which pollinators, herbivores and plants coexist. Since the justification of their choice for the type II is rather weak, I want to know how sensitive the main finding of this study (herbivore induced pollinator limitation promotes persistence) is to the choice of the response type. I would recommend that they take the general form $1/(1+ch^b)$ as parameter b can be included in the analysis. My guess would be that when $b > 2$ the range under which community persistence occurs will be smaller because pollinator rates do not decline very rapidly at low levels of herbivory. For this reason herbivore densities will decrease less rapidly which could drive pollinators to extinction.

All of the functional responses are tested and concerns are addressed in the response to Reviewer 2's comments. Please note the consistency of the rescue effect of HIPL under the Type I, Type III, and Mixed Saturating functional response forms. We have compiled all of these analyses into the new Supplementary Info C, copied further below.

Here we further address the specifics of Reviewer 3's comments regarding the Mixed Saturating FR:

Reviewer 3 made a reasonable assumption that $b > 2$, or $b > 1$ for that matter, would reduce the range of the rescue effect and community persistence in the parameter space. The Reviewer thought that when $b > 2$, using the Mixed Saturating model, the lack of HIPL at lower levels of herbivory could drive the pollinator to extinction due a reduced level of indirect control over the herbivore. Once our university computing core finally (and briefly) came back online in a functional manner, we tested where persistence occurred across $\{c, b, r_H\}$ parameter space. This was a large parameter sweep involving simulations over 453,000 parameter combinations. While the hypothesis is sensible, we found the exact opposite was the case when we tested this with the model. In other words, $b > 1$ expands the range under which persistence occurs.

The first indication of this can be seen in Review Figures 3 and 4 (please see response to Reviewer 2's comments above). In Review Fig 3, higher levels of b did not reduce the range under which community persistence occurred at low levels of r_H (herbivore attack rate). Review Figure 4 then shows that higher values of b actually allow for a *lower* required value of c to sustain persistent communities, at least for one higher value of r_H . If that is the case, then we should see a *greater* range of $\{r_H, c\}$ parameter space supporting persistence as values of b increase. In fact, this is what we see (Review Figure 12). Review Figure 12 measures the number of $\{r_H, c\}$ parameter combinations which supports the persistence the community through the rescue effect. The results in Review Figure 4 (above) show this increase is mainly driven by a lowering in the value of c required to support the system.

Review Figure 12: Measurements of the $\{r_H, c\}$ parameter space area across a range of b values where the HIPL rescue effect enables persistent communities. This shows the range of the rescue effect increases with higher values of b . Parameter values are $b_F = 1.45, b_P = 1.55, c_{FH} = 0.7, d_F = 0.1, d_P = 0.1, d_H = 0.1, h = 1.1, a = 0.1$.

Additionally we can plot r_H against b . Similar to the relationship between r_H and c , (though less dramatic) increases in the value of b increases the range of r_H values which still create persistent systems (Review Figure 13 below). Instead of showing values of F , the heatmap element of Review Figure 13 shows the lowest value of c under which the system persists at different combinations of r_H and b through the HIPL rescue effect. The lower the value of c , the larger the range of community persistence. Intuitively, lower values of r_H require lower values of c for the rescue effect. Higher values of b also can also decrease the minimum value of c needed to support persistence. This is shown in two color schemes in Review Figure 13 to make the point clearer.

Lowest Value of c Which Supports Persistent Communities Across the Range of all Tested Values of r_H and b

Review Figure 13: Heatmap showing the minimum value of c required to support system persistence through the rescue effect of HIPL. The same figure is shown with two different color schemes to represent the minimum required c value. Parameter values are $b_F = 1.45$, $b_P = 1.55$, $c_{FH} = 0.7$, $d_F = 0.1$, $d_P = 0.1$, $d_H = 0.1$, $h = 1.1$, $a = 0.1$.

As for using the Mixed Saturating form ($1/(1+ch^b)$), there are number of reasons we feel the ($1/(1+ch)$) Type II form is more suitable for the main paper's analysis and presentation. First, the AICc weight better supports the Type II form in both the binned and non-binned versions of Kessler's data. The Mixed Saturating form does not have comparable support. Second, the estimated value of b is 1.196. This is much closer to 1 than 2, and such a small change in the shape of the function will have little effect on the overall qualitative result shown in the main paper. Third, inclusion of an additional analysis across the full range of b parameter values adds a rather sizeable amount of content to the main paper (as seen above and in Supplementary Information C). Finally, this additional content is described in the necessary detail in the new Supplementary Material C (please see below). Therefore, we put forth that the Type II analysis is a better suited for description in the main paper while the further analysis best belongs in the Supplementary Info.

3) Thus the statistical support for a type II response is weak and it is unclear to what extent the main result is affected by this choice. Yet, it seems to be the key selling point of the paper: functional reductions in pollination services and their surprising potential in mutualist communities. Please provide evidence that this finding is robust to the assumptions made.

Please see the above responses.

Also, for ease of reading, we are adding the newly created Supplementary Information C which describes, in detail, the analysis of the other functional responses.

We inform the reader of Supplementary Info C on **Line 458**:

“However, additional analysis indicates that the other functional forms of HIPL can consistently indirectly control herbivore population growth when used in Equation 1 (Supplementary Info C).”

Please find Supplementary Info C just below:

///New Addition, Supplementary Information C

Supplementary Info C – HIPL driven rescue effect with other functional responses:

While we argue that the Type II functional response has the most support in the available data, it is possible that other pollination systems may support a different functional form. Therefore, there is merit in investigating the consistency of the rescue effect provided through HIPL when using other functional responses in the model. In other words, here we will test the potential for the rescue effect with the Type I, Type III, Mixed Saturating, and Concave functional responses. Analysis shows that the rescue effect can be readily replicated across all functional response forms, with only the Concave response showing a noticeable reduction in the range of parameter space supporting community persistence.

Supplementary Info C.1 – Type I HIPL Functional Response

The Type I functional response for HIPL is the linear equation:

$$v(c, H) = 1 - c * H$$

We can integrate in form of $v(c, H)$ into the model using the Piecewise function:

$$v(c, H) = \begin{cases} 1 - c * H & \text{when } 1 > c * H \\ 0 & \text{when } 1 \leq c * H \end{cases}$$

The Piecewise formulation stops $v(c, H)$ from becoming negative at any time in simulations. This formulation means that $v(c, H)$ decreases linearly with increased herbivore abundance (H) until it reaches 0. The value of $v(c, H)$ then remains at 0 when $H \geq 1/c$. As in the main paper, the interaction rate of pollinators and flowering plants is assumed to be 1 when herbivore abundance and damage is zero. With this instantiation of the model, the linear Type I HIPL functional response can still produce the rescue effect. It is possible to create similar bifurcation heatmaps as shown in Figure 3 in the main paper to illustrate this result (Supplementary Fig C1).

Value of F in Asymptotic Behavior of Type I HIPL Model

Supplementary Figure C1: A two-dimensional bifurcation heatmap showing the abundance of F (flowering plant) in the asymptotic behavior of the model using a TYPE I functional response for HIPL. Different asymptotic behaviors of the model are shown as different colors across the $\{r_H, c\}$ parameter space. Where parameter combinations create stable equilibria, F abundance is shown in the green color scale. Where values create stable limit cycles, F abundance is shown in the sunset color scale. Areas in white represent herbivore driven local extinction. $r_F = 0$; $b_F = 1.265$; $b_P = 1.4$; $c_{FH} = 0.7$; $d_F = 0.2$; $d_H = 0.25$; $d_P = 0.2$; $h_F = h_P = 1.1$; $\alpha = 0.1$.

Supplementary Info C.2 – Type III HIPL Functional Response

Similar to the Type I and Type II functional response, the Type III form of HIPL was also found to produce the rescue effect. In this case, $v(c, H) = \frac{1}{1+cH^2}$. Again we present the results in the bifurcation heatmap figure (similar to Fig 3 in the main paper). The Type III functional response can allow for the rescue effect over similarly large subset of the parameter space (Supplementary Figure C2).

Value of F in Asymptotic Behavior of Type III HIPL Model

Supplementary Figure C2: A two-dimensional bifurcation heatmap showing the abundance of F (flowering plant) in the asymptotic behavior of the model using a TYPE I functional response for HIPL. Different asymptotic behaviors of the model are shown as different colors across the $\{r_H, c\}$ parameter space. Where parameter combinations create stable equilibria, F abundance is shown in the green color scale. Where values create stable limit cycles, F abundance is shown in the sunset color scale. Areas in white represent herbivore driven local extinction. $r_F = 0$; $b_F = 1.465$; $b_P = 1.615$; $c_{FH} = 0.7$; $d_F = 0.2$; $d_H = 0.25$; $d_P = 0.2$; $h_F = h_P = 1.1$; $\alpha = 0.1$.

Supplementary Info C.3 – Mixed Saturating HIPL Functional Response

The Type III functional response is actually a subset/subcase of the Mixed Saturating form. From a modeling standpoint (both statistical and dynamic), the Mixed Saturating Case is a more complicated case because there are three parameters to test ($\{r_H, c, b\}$). Regardless, the Mixed Saturating form can produce the rescue effect result described in the main paper, but the details are more involved. The Type I, II, III functional responses only had one parameter per function (c), so it was possible to make the 2-D

bifurcation heatmaps. In this case, there is more than one parameter for the mixed saturating functional response (parameter c and parameter b):

$$v(c, b, H) = \frac{1}{1 + cH^b}$$

Therefore, the previous 2-D $\{r_H, c\}$ bifurcation heatmaps do not show all the details and we will need to show multiple figures to describe the full dynamics. In this model we are actively changing the values of b for the first time, so we initially parse through values of b to test their comparative effects. The values of b will vary from 1 to 3 allowing us to compare dynamics of a Type II response ($b = 1$) with a Type III ($b = 2$) and the Mixed Saturating case (b generally greater than 1). First, testing the effect of b on a lower interacting system reveals that higher values of b do not restrict the range of community persistence (Suppl Fig C3a). In fact, on the lower values of c , it appears higher b values allow for community persistence. Testing a more interactive system (Suppl Fig C3b) offers clearer support for this idea.

Supplementary Figure C3: a) Two-dimensional bifurcation heatmaps showing the abundance of F (flowering plant) in the asymptotic behavior of the model using a Mixed Saturating functional response for HIPL. Different asymptotic behaviors of the model are shown as different colors across the $\{b, c\}$ parameter space. Where parameter combinations create stable equilibria, F abundance is shown in the green color scale. Where values create stable limit cycles, F abundance is shown in the sunset color scale. Areas in white represent herbivore driven local extinction. a) $b_F = b_P = 0.78$, $r_H = 0.67$, $c_{FH} = 0.7$; $d_F = 0.2$; $d_H = 0.25$; $d_P = 0.2$; $h_F = h_P = 1.1$; $\alpha = 0.1$. b) $b_F = 1.45$, $b_P = 1.55$, $r_H = 0.91$, $c_{FH} = 0.7$; $d_F = 0.2$; $d_H = 0.25$; $d_P = 0.2$; $h_F = h_P = 1.1$; $\alpha = 0.1$.

To more fully understand the role of parameter b in model dynamics and the rescue effect, we expanded the parameter sweep to include r_H so that persistence could be measured across over 453,000 parameter combinations in $\{c, b, r_H\}$ parameter space. The parameter sweep done to construct this graph was done with r_H values from 0.45 to 1.0 with 0.01 steps, c values from 0.0 to 2.0 with 0.02 steps, and b values from 1 to 3 with 0.02 steps. Upon completion of the analysis, we found that increased values of b expand the range of the HIPL derived rescue effect in $\{r_H, c\}$ parameter space (Suppl Fig C4). As indicated in Suppl Fig C3b, this expansion largely results from the reduction in the value of c required to sufficiently control the herbivore population prompting the rescue effect.

Supplementary Figure C4: Measurements of the number of $\{r_H, c\}$ parameter combinations across a range of b values where the HIPL rescue effect enables persistent communities. The y-axis in the case is the actual count of distinct parameter combinations which HIPL supports persistent communities. This shows the range of the rescue effect increases with higher values of b . Parameter values are $b_F = 1.45, b_P = 1.55, c_{FH} = 0.7, d_F = 0.1, d_P = 0.1, d_H = 0.1, h = 1.1, \alpha = 0.1$.

Higher values of b not only reduce the level of c required for persistence, they also slightly increase the level of herbivore attack rate (i.e. higher r_H) that the system can withstand before local extinction (Suppl Fig C5). The heatmap in Supplementary Figure C5 does not show asymptotic value of the flowering plant population as it does in other heatmaps. Instead it shows the lowest value of c (lowest level of HIPL) that the system could withstand and still persist. Intuitively, lower values of r_H require lower values of c for the rescue effect. Higher values of b also can also decrease the minimum value of c needed to support community persistence. This is shown in two color schemes in Supplementary Figure C5 to make the point clearer.

Supplementary Figure C5: Heatmap showing the minimum value of c required to support system persistence through the rescue effect of HIPL. This minimum value of c is shown in the colors of each graph explained by the color legend to the right of each figure. The same figure is shown with two different color schemes to represent the minimum required c value in an easily visible manner. Parameter values are $b_F = 1.45, b_P = 1.55, c_{FH} = 0.7, d_F = 0.1, d_P = 0.1, d_H = 0.1, h = 1.1, \alpha = 0.1$.

The cause of this increase in the range of community persistence across parameter space results from the fact that higher b values (i.e. $b > 1$) do two things to the shape of the $v(c, H)$ function. First, it creates a delay in the immediate effect of HIPL, such that higher herbivore abundance is required to see a decrease in pollinator visitation ($v(c, b, H)$). Second, once the herbivores are abundant, higher b values make the resulting reduction in pollinator visitation progressively steeper, such that the decline in pollinator visitation is quite rapid. While the first effect would seem detrimental, when coupled with the second effect, it can actually be beneficial. A less immediate decline in $v(c, b, H)$ at low herbivore abundance can actually help the pollinator and plant populations rebound during troughs in the population trajectory because low herbivore abundance won't impede pollination. This effect by itself would then fail to control herbivore populations as the populations rebounded, but the concurrent steep decline in pollination once herbivore abundance become sufficiently high (the second effect above), helps regain the indirect control of the herbivore population growth through greater reductions in v (Supplementary Figure C6).

Supplementary Figure C6: The resultant value of the pollination visitation parameter $v(c, b, H)$ (shown in blue) across herbivore abundance (H) at different levels of the parameter b when using the Mixed Saturating functional response form: a) $b = 1$, b) $b = 2.2$, c) $b = 3.52$. The red dashed line shows the value of $v(c, b, H)$ when the herbivore abundance (H) equals 2.0. Here, $c = 0.34$. As the value of b increases, the shape of the function changes such that the eventual decrease in $v(c, b, H)$ becomes very steep. Therefore, $v(0.34, b, 2)$ decreases resulting in more HIPL and lower pollination rates.

Supplementary Info C.4 – Concave HIPL Functional Response

Various numerical simulations show that it is possible to recreate the rescue effect with the Concave Functional Response (Supplementary Figure C7). However, the Concave functional response generally created the smallest parameter space in which the rescue effect could be found. By creating the longest delays in declining the pollinator visitation rate (v), the Concave functional response can significantly hinder any possible direct control of the herbivore population through HIPL. The concave model is the least supported direct curve fit we attempted, so we claim that the only functional response type that noticeably reduces the range of the rescue effect in the model does not seem well supported. With this we can claim that the main results presented in the paper are robust to most functional responses types.

Review Figure 5: Three time series showing the rescue effect using a Concave functional form of HIPL. Parameter values: $r_H = .58$, $b_F = 1.095$, $b_P = 1.095$, $c_{FH} = 0.7$, $h = 1.1$, $c = 0.13, 0.338, 0.868$, $\alpha = 0.1$, $d_F = 0.1$, $d_P = 0.1$, $d_H = 0.1$. The green line, orange line, and black line represent the flowering plant, pollinator, and herbivore respectively. The pink line is the value of the $v(c, H)$ function as a response to the herbivore abundance.

///End of New Addition, Supplementary Information
C

Methods:

1. Is the model (Eq. 1) the same as is provided in Jang (2002 JMatBiol 44(2) 129-149) with the exception that $v(c, h)$ is specified to be a type II? If not, please indicate how it differs from this model. If another model was used as basis, please mention that one,

Beyond some further deconstruction of single parameters into component parts, the major difference is the $v(c, h)$ inclusion and quantitative exploration. Both of our models mainly build off of classical consumer-resource equations modified into mutualisms in an overarching 3-dimensional system. We describe the distinctions in lines 154-175.

2. The authors do not provide units for the parameters and the state variables. This makes it very hard to check the model and interpret the parameters and the parameter values.

As is the case with models of this type, parameters provide proportional per capita rates per unit time step. So writing out the units would be something like per individual per unit time, Writing it out would look something like ($ind^{-1}t^{-1}$). Handling time can be considered a unit of time, but it comes into play as part of a rate of attack and conversion. As for the state variables, there is no SI unit for organismal biomass. So, the variables themselves can denote population numbers or concentrations (number per area) or some other scaled measure of the population's sizes. Whatever form it takes would become the individual unit. The exact definition can be determined when model output is compared to data that the equations can model.

Further info can be found in the below publication:

H. I. Freedman, *Deterministic Mathematical Models in Population Ecology*. New York: Marcel Dekker, 1980.

These explicit statements are often not included in papers, but you can see an example of it written out in the George and Loueille 2014 citation:

Georgelin, E. & Loueille, N. Dynamics of coupled mutualistic and antagonistic interactions, and their implications for ecosystem management. *Journal of Theoretical Biology* 346, 67–74 (2014).

We added the following to the Table heading of Table 2: **Line 645:**

“Parameters are measured per individual per unit time.”

3. How were the equations solved and which integration routine was used?

We avoided Euler's method. Using Mathematica, these ODEs were solved using the function NDSolve using Explicit RungeKutta methods. By using a small step size in integration, the computing time required to solve single simulations was somewhat larger, but provided better precision in the numerical approximation trajectories.

We added this to the methods section

Line 266:

“Analysis was done through Mathematica 10 using NDSolve with Explicit RungeKutta methods. Large scale analysis was facilitated by University of Michigan's FLUX computing core.”

4. Parametrisation. How were the parameter values chosen? From fig. B1 I understand the herbivores occur in highest abundances, followed by pollinators, and finally plants. This seems strange to me. Or are the abundances of the herbivores, pollinators and plants expressed in different units? How does the function $v(c,h)$ look like when applying to the herbivore abundances that are typically found in the model (for some representative values of c)? In other words, is the scaled slope comparable to the scaled slope that was found in the field?

Parameter values were generally within ranges used by other theoretical studies cited in this work. There is not widely agreed upon values for some of these parameters, which is the reason why we did parameter sweeps to test multiple values. The variables are not expressed in different units. The order of abundance rank is certainly feasible depending on the herbivores. Soft bodied insects like thrips, aphids, scale insects, etc. can easily reach very high abundances on a single plant alone. On the other hand, plants (depending on the species) can have multiple flowers per individual to support an intermediate abundance of pollinators.

Also, dependent on the specific parameter values the abundances ranks can be easily be changed while keeping the core dynamic of HIPL indirectly limiting herbivores and “rescuing” the system. For some quick examples, see below:

Review Figure 14: Here we see two examples of the core dynamic of the paper shown with different levels of abundance due to different parameter values. As the parameter c increases, the system can go from local extinctions to oscillations and stable dynamics. Parameter values are $r_H = .68, b_F = .73, b_P = .745, c_{HF} = .7, d_F = d_P = 0.1, d_H = .25, h_H = h_P = 1.1$

Review Figure 15: $r_H = .71, b_F = .92, b_P = .47, c_{HF} = .7, d_F = d_P = 0.1, d_H = .25, h_H = h_P = 1.1$.

5. The parameter value for cfh is missing in the figures in the main text.

Apologies, it has been added.

6. How is it possible that below r_H values of 0.55 populations go extinct? I would guess that at low values of r_H herbivores would go extinct, but not F and P?

The Reviewer raises a good point here about problems with our wording in the description of Fig 3. They are correct in their point that herbivores go extinct, but certainly not F and P when $r_H < 0.55$. The data displayed in Figure 3 only shows results at parameter location where the entire 3-variables (F,H, and P) persist. Other dynamics are simple, either system goes fully extinct (the lower right part of the figure) or H goes extinct and F and P go to carrying capacity.

We have changed the wording in the figure legend to make this clearer:

Line 671:

“Where parameter combinations create stable equilibria, F abundance is shown in the green color scale. Where values create stable limit cycles, F abundance is shown in the sunset color scale. The switch between the two color schemes represents the Hopf bifurcation shown in Fig 2. Values which lead to either extinction of H ($r_H < 0.55$) or full system extinction (lower right portion of figure) are shown in white.”

7. Fig 5. b,c I have difficulties understanding those graphs.

We assume the reviewer meant Fig 4 b,c in this case, as there is no Figure 5. Apologies if we are not addressing the intended issue.

Figure 4 b and c show the important dynamics which drive the pattern shown in the Figure 4a.

In Fig 4b:

When $r_F < 0.7$ and the system supports stable trajectories, the equilibrium values of variables change as r_F increases from 0. In particular, the herbivore equilibrium H^* increases and the pollinator

equilibrium (P^*) is driven lower and lower until it eventually reaches 0. Fig 4a shows the increase in H^* and the decrease in P^* for the different values of r_F .

In Fig 4c:

Eventually, with sufficiently high values of r_F the equilibria become unstable and the system/populations begins to oscillate through time. This creates peaks and troughs in each population's trajectory. As r_F increases, the amplitude of these oscillations increases, meaning higher peaks and lower valleys/troughs (this is displayed in the bifurcation diagram of H shown in black). Lower trough values in the populations oscillations take longer to recover from, so the amount of time across the simulation that each population spends at lower values increases with increased r_F .

Herbivores also experience these oscillations and as the amplitude increases, the time herbivores spent at low abundances increases. We show this with the red line measuring the number of time steps H is below 0.5 during oscillations with increasing values of r_F . The value of 0.5 and this analysis are a heuristic measures for this dynamic but they describe the process well.

This longer period of time with lower abundances of H is what allows the population of pollinators (P) to rebound and begin oscillating at higher values (shown in Fig 4a). You can see that the bifurcation which splits the overall system dynamic from stable equilibria to stable oscillations ($r_F \approx .8$ in Fig 4c), occurs before the pollinator population can rebound into appreciable abundances ($r_F \approx 1.0$). Only upon oscillations reaching amplitudes where H is at low abundances for an extended period of time, can the pollinator oscillations again reach sustainable levels ($r_F \approx 1.0$ in both Figure 4a and 4c).

We added the following in an effort to increase the clarity of this description:

Line 378:

“These large oscillations create higher peaks in H but consequently result in lower minima values (bifurcation diagram of H in Fig 4c Fig 4c). Lower minima values mean longer recovery times from low population abundances. This result in longer periods of time where herbivore abundance is low in-between oscillatory population peaks. Heuristically, this can be shown by measuring the amount of time $H < 0.5$ as r_F increases (Fig 4c, red line).”

Line 683: In the legend for Figure 4:

As r_F increases past 0.8, the system and H populations begin to oscillate with increasing amplitude.

Reviewers' comments:

Reviewer #1 confidentially expressed satisfaction with the revision

Reviewer #2 (Remarks to the Author):

The authors responded satisfactorily to my comments

Reviewer #3 (Remarks to the Author):

Dear Editor,

The authors have done a great job of revising the manuscript and arguing their case. I enjoyed the open-minded and diligent consideration of my comments, and I was impressed by the great number of simulations they did to make their point.

My main concern was that the outcomes of the model were sensitive to the choice of the functional form (either type I, II, III). The authors show that persistence of herbivores, pollinators and plants is rather independent of the functional form that is chosen (only the concave form really limits the parameter space in which persistence is possible). I agree to only present the results of the $b=1$ simulations (Type II), and that the other simulations belong to the supplementary material. My initial hypothesis that when b becomes larger, the range in which the rescue effect exists will be reduced was not supported by the simulations. This is an interesting finding and it is probably worth mentioning a summary of this result in the discussion. In addition, I would like to see this control analysis (playing with values of b) mentioned earlier on in the paper (results; in words, no figures), as the paper gets stronger by the fact that the results are not very sensitive to the choice of b , and because more readers will reason in the same way as my colleague-reviewer and I did.

About the statistical support of the type II functional response. I am happy that the wording now better matches the statistical findings. I do not agree with that the binning procedure better matches with the mean field assumption that the model takes. I agree that the measurement of an individual plant will not be representative because of the complicating factors are mentioned. However, the expected value for multiple individuals is representative (that is also what the binned approach assumes) and would represent the mean field approach. I would say that a binned approach is only justified when there is a (large) uncertainty in the quantification of damage. In addition, I do not agree that the binned approach better communicates the idea of herbivore induces the decline in visitation rates. If the authors want to stick to the binned approach that is fine with me.

My comment on "how did the authors include the standard errors around the average in their statistical model" was interpreted in a different way than what I intended, but it lead to interesting simulations. I agree that including uncertainty in the model would be a separate paper and would require a more thorough analysis. What I meant is that in the statistics on the binned data, the authors do not use the standard error of the means to be included in the estimation of, and comparison of the curves. Rather, the authors treat the mean percentage of flowers in each bin as it was measured with the same precision. Alternatively, they could have given weights to the mean in each bin by taken into account the number of observations that was underlying each binned mean. This can be done, through applying "weights" in the function 'nls' in R that the authors used. My suspicion is that the results, in terms of which type of functional response best fits the data, become more comparable to the ranking of models on the raw data. Implementing weights is not much work, I propose that the authors carry out this analysis and when the outcome changes substantially they can adjust their wording accordingly (see attached R script).

With respect to the curve fitting on the raw data. I am surprised to see that the AIC of the Poisson

glm is so much higher than the AIC when assuming a normal distribution – something that I cannot readily explain. I agree with the comment of your statistician that applying a non-linear function in combination with a log-link would be a bit strange. However, a log-link is not obligatory; it is applied by default to always get positive values for the predicted mean (which is required by the Poisson). I would have gone about this as follows. Take a glm with a poisson distribution, and/or a negative binomial, choose the same functions that were tested before (type I, II, III etc) and choose an “identity” link instead of a log link. My experiences is that by playing choosing proper starting values, a good model can be found (thus `glm(...,poisson(link = "identity"))` or `glm.nb` in R). Given that the authors strongly prefer the binned approach, I would suggest to continue the analysis considering the weights in the nls.

About the naming of the functional forms (type I, II and III): I leave that to the authors.

Dear Reviewers,

Thank you for your thoughts and comments. We are glad to see that the reviewers were generally happy with the changes we made to the earlier version of the manuscript and would like to voice our appreciation for their thoughtful inputs.

Below we addressed all the remaining concerns by Reviewer 3, whose comments were particularly thoughtful and helpful. These additional considerations helped to clarify the manuscript further, hopefully to the advantage of the readers.

Reviewers' comments:

Reviewer #1 confidentially expressed satisfaction with the revision

Reviewer #2 (Remarks to the Author):

The authors responded satisfactorily to my comments

Reviewer #3 (Remarks to the Author):

Dear Editor,

The authors have done a great job of revising the manuscript and arguing their case. I enjoyed the open-minded and diligent consideration of my comments, and I was impressed by the great number of simulations they did to make their point.

My main concern was that the outcomes of the model were sensitive to the choice of the functional form (either type I, II, III). The authors show that persistence of herbivores, pollinators and plants is rather independent of the functional form that is chosen (only the concave form really limits the parameter space in which persistence is possible). I agree to only present the results of the $b=1$ simulations (Type II), and that the other simulations belong to the supplementary material. My initial hypothesis that when b becomes larger, the range in which the rescue effect exists will be reduced was not supported by the simulations. This is an interesting finding and it is probably worth mentioning a summary of this result in the discussion. In addition, I would like to see this control analysis (playing with values of b) mentioned earlier on in the paper (results; in words, no figures), as the paper gets stronger by the fact that the results are not very sensitive to the choice of b , and because more readers will reason in the same way as my colleague-reviewer and I did.

We agree that it is likely that other readers will reason similarly to Reviewer 2 and Reviewer 3. In accordance with the Reviewer's suggestions, we have added an earlier description of these results:

First, line 189, we explicitly introduce the non-integer exponent $b > 1$:

“The Mixed Saturating model tests the effect of a response model with a scalar multiplier, c , on h and a potential non-integer exponent, b (Table 1).”

Then on line 204:

“Given the appreciable support for the Mixed Saturating form, we did analyze cases where $b > 1$. We also analyzed the effects of the other functional forms of HIPL. Analysis showed consistent results with those presented here (see Discussion).”

Finally, starting on line 459 in the Discussion, we added a more detailed discussion of the results when other functional forms are used. This includes a description of the results when $b > 1$ in the Mixed Saturating case. We needed to make a few further changes in sentence order to preserve the general flow of ideas. It now reads:

“The HIPL displayed in these (Barber’s) data did not originate from HI-VOCs, but from direct physical effects of herbivory on flower attractiveness and mycorrhizal fungi colonization. Perhaps other mechanisms of visitation reduction may be prone to different functional forms. Additionally, the Barber et al study system was a less specialized pollination system, and the two major pollinators were both well-known generalists (bumble bees and honey bees). This may also affect the functional form of HIPL and indicates there is a need to study these effects in more pollination mutualisms along the full gradient of specialization to generalization.

Prompted by the possibility of other functional forms, we analyzed model dynamics using alternate functions for $v(c, H)$. Overall, these analyses show that other functional forms can consistently indirectly control herbivore population growth when used in Equation 1 (Supplementary Info C). Only the Concave functional form was found to noticeably limit the range of community persistence in tested parameter space. This occurred because the Concave function leads to long delays in the reduction of pollination services until herbivores reach comparatively high abundances, consequently, eliminating the indirect control of herbivore population growth. It is reasonable to assume that such a dynamic would also occur in the Mixed Saturating case when $b > 1$. While there is a similar delay in HIPL when $b > 1$, it is relatively limited and is followed by such a steep decline pollinator visitation that the effective indirect control of herbivore populations can occur at lower values of c as the value of b increases (Supplementary Info C.3).”

About the statistical support of the type II functional response. I am happy that the wording now better matches the statistical findings. I do not agree with that the binning procedure better matches with the mean field assumption that the model takes. I agree that the measurement of an individual plant will not be representative because of the complicating factors are mentioned. However, the expected value for multiple individuals is representative (that is also what the binned approach assumes) and would represent the mean field approach. I would say that a binned approach is only justified when there is a (large) uncertainty in the quantification of damage. In addition, I do not agree that the binned approach better communicates the idea of herbivore induces the decline in visitation rates. If the authors want to stick to the binned approach that is fine with me.

Given the consent of Reviewer 3 and given that the other Reviewers were content with the binned approach we used, we would prefer to use the binned approach. However, we did consider Reviewer 3’s comments on using weights in the analysis below.

My comment on “how did the authors include the standard errors around the average in their statistical model” was interpreted in a different way than what I intended, but it lead to interesting simulations. I agree that including uncertainty in the model would be a separate paper and would require a more thorough analysis. What I meant is that in the statistics on the binned data, the authors do no use the standard error of the means to be included in the estimation of, and comparison of the curves. Rather, the authors treat the mean percentage of flowers in each bin as it was measured with the same precision. Alternatively, they could have given weights to the mean in each bin by taken into account the number of observations that was underlying each binned mean. This can be done, through applying “weights” in the function ‘nls’ in R that the authors used. My suspicion is that the results, in terms of which type of functional response best fits the data, become more comparable to the ranking of models on the raw data. Implementing weights is not much work, I propose that the authors carry out this analysis and when the outcome changes substantially they can adjust their wording accordingly (see attached R script).

Given the use of binned data, we agree this could be considered a reasonable alternative approach. In line with the suggestion of the reviewer, we added the weights to the nls regressions and ran the analysis to test the effects. We thank the Reviewer for providing code to aid in the consideration of weights and help us understand their thought process/suggestion.

Reviewer 3 mentioned that this might change the results, but the reason why Reviewer 3 got different AICc values than we did is that Reviewer 3 created the x-axis differently than we did. Looking

at their provided code, one can see that they obtained the x axis values by creating bins and labeling them numerically 0.1, 0.2, 0.3,...1,0 corresponding to their bin. Their R code is below:

```
x = c(0:10)/10 ##This creates the numerically labeled x-axis.
plot(out ~ x)
nls.1 = nls(out ~ j/(1+c*x),start=list(j=0.1,c=-0.09))
nls.1w = nls(out ~ j/(1+c*x),start=list(j=0.1,c=-0.09),weights=as.vector(no),data=data)
```

In the paper, we state that the x-axis values assigned to each bin were drawn from the mean damage in that bin. Please see line 176, copied below from the submission:

“In order to ascertain the functional form of this negative correlation, Kessler et al’s data¹⁸ has been broken into 11 sets. The first set (serving as the control) measures average pollinator visitation at 0% herbivore damage and is followed by 10 categories each grouped by taking the averages of herbivore damage and pollinator visitation in 10 percentage point steps (Figure 1). This results in 11 averaged data points with standard errors on the x (herbivore damage) and y (pollination percentage) axes (Figure 1).”

One can also see the horizontal damage error bars in Figure 1. The means better represent the values of those bins than just numeric labels. Using the x-axis described in the paper, one can see that the results do not change substantially and the main result remains the same, the Type II form is still the best choice (see Review Table 1). Though it does affect the relative AICc weights so that the Type II weight is somewhat lower and the second best model choice is now the Type I/Linear form, while the Mixed Saturating model is the 3rd best choice. However, as stated above, the main result is consistent.

Fitted models to averages	Example of functional form	Estimated Parameters	Significance and Fit	AICc	AICc weight
1.)Type I/Linear: $\sim ch + i$		$c = -0.654,$ $i = 0.864$	$p = (c)4.13e^{-6}$ $(i)8.43e^{-10}$ $R^2 = 0.911$	-315.09	0.215
2.)Type II: $\sim \frac{i}{1 + ch}$		$c = 1.807,$ $i = 0.953$	$p = (c)2.77e^{-5}$ $(i)1.89e^{-9}$	-317.10	0.587
3.)Type III: $\sim \frac{i}{1 + ch^2}$		$c = 2.491$ $i = 0.838$	$p = (c)3.57e^{-4}$ $(i)2.86e^{-9}$	-312.59	0.062
4.)Mixed Saturating: $\sim \frac{i}{1 + ch^b}$		$c = 2.000$ $b = 1.243$ $i = 0.920$	$p = (c)1.13e^{-4}$ $(b)0.242e^{-4}$ $(i)2.62e^{-8}$	-313.90	0.119
5.)Concave: $\sim i * \left(1 - \frac{h}{100}\right)^c$		$c = 0.606,$ $i = 0.842$	$p = (c)2.58e^{-4}$ $(i)2.36e^{-8}$	-310.16	0.018

Review Table 1: Results from the curve fitting analysis using the weights as suggested by Reviewer 3. One can see that the best fit model is still the Type II form.

Due to the lack of any substantial changes, the similar major result, and Reviewer 3's interest in the Mixed Saturating model, we decided to present these results in the Supplementary Info A.3. That way the weighted analysis is acknowledged and available, but the secondary focus on the interesting results from the Mixed Saturating model is not lost in the main paper.

As such, the following changes were made:

On **line 199**, we added:

“Furthermore, incorporating standard error into the nls regression and giving weights to each mean value also finds the Type II response to be the best supported form (Supplementary Info A.3).”

Then in **Supplementary Information A.3** we added the following:

Supplementary Info A.3 – Fitting binned data from Kessler et al 2011 with weighted bins

In using the binned data, it is reasonable to also consider the effects of standard errors of the mean values from the bins in estimation and comparison of the curves. This can be done by counting the number of observations per bin and using them as weights with the “weights” argument provided in the nls function in R. When we do so, we see no change in the main result, the Type II functional form is still the best supported model from the analysis (Supplementary Table A.3). However, there is more comparable support for the Type I/Linear form and still appreciable support for the Mixed Saturating form. These results provided additional prompting to study the effects of the other functional forms on the model results (see Supplementary Info C).

Supplementary Table A.3: Table describing the results of the curve fitting to the 5 candidate response models: Type I/Linear, Type II, Type III, Mixed Saturating, Concave when including the weights in each bin. Here h represents the level of herbivory. The parameters c and b determine the shape of the curve and i is the intercept. Equation representations of each model are given along with a pictorial example of each model. Estimated parameters that are significant have their p values bolded. The Type II functional response has the highest Akaike Information Criterion weight of 0.587.

Fitted models to averages	Example of functional form	Estimated Parameters	Significance and Fit	AICc	AICc weight
1.)Type I/Linear: $\sim ch + i$		$c = -0.654,$ $i = 0.864$	$p = (c)4.13e^{-6}$ $(i)8.43e^{-10}$ $R^2 = 0.911$	-315.09	0.215
2.)Type II: $\sim \frac{i}{1 + ch}$		$c = 1.807,$ $i = 0.953$	$p = (c)2.77e^{-5}$ $(i)1.89e^{-9}$	-317.10	0.587
3.)Type III: $\sim \frac{i}{1 + ch^2}$		$c = 2.491$ $i = 0.838$	$p = (c)3.57e^{-4}$ $(i)2.86e^{-9}$	-312.59	0.062
4.)Mixed Saturating: $\sim \frac{i}{1 + ch^b}$		$c = 2.000$ $b = 1.243$ $i = 0.920$	$p = (c)1.13e^{-4}$ $(b)0.242e^{-4}$ $(i)2.62e^{-8}$	-313.90	0.119
5.)Concave: $\sim i * \left(1 - \frac{h}{100}\right)^c$		$c = 0.606,$ $i = 0.842$	$p = (c)2.58e^{-4}$ $(i)2.36e^{-8}$	-310.16	0.018

With respect to the curve fitting on the raw data. I am surprised to see that the AIC of the Poisson glm is so much higher than the AIC when assuming a normal distribution – something that I cannot readily explain. I agree with the comment of your statistician that applying a non-linear function in combination with a log-link would be a bit strange. However, a log-link is not obligatory; it is applied by default to always get positive values for the predicted mean (which is required by the Poisson). I would have gone about this as follows. Take a glm with a poisson distribution, and/or a negative binomial, choose the same functions that were tested before (type I, II, III etc) and choose an “identity” link instead of a log link. My experiences is that by playing choosing proper starting values, a good model can be found (thus `glm(.....,poisson(link = "identity"))` or `glm.nb` in R). Given that the authors strongly prefer the binned approach, I would suggest to continue the analysis considering the weights in the nls.

As per Reviewer 3's suggestion and the other Reviewer's acceptance of our binned approach, we will continue the binned analysis.

About the naming of the functional forms (type I, II and III): I leave that to the authors.

Given that we define the forms in the paper and the other Reviewer's acceptance of the terms, we will keep the terms in their current form unless the Editor has strong opinions one way or the other.

REVIEWERS' COMMENTS:

Reviewer #4 (Remarks to the Author):

Dear Editor,

I am satisfied with the response of the authors to my comments.

Response to Reviewer/Referee Comments:

Reviewer 4 only stated:

“Dear Editor, I am satisfied with the response of the authors to my comments.”

No further issues were raised by the Reviewers. We endeavored to address all Reviewer comments throughout the review process so we are happy to hear that the Reviewers are satisfied with our changes.

-Paul Glaum and Andre Kessler